



# Ferrihydrite associated organic matter (OM) stimulates reduction by *Shewanella oneidensis* MR-1 and a complex microbial consortia

Rebecca E. Cooper[1], Karin Eusterhues[2], Carl-Eric Wegner[1], Kai Uwe Totsche[2], Kirsten Küsel[1,3]

[1]Aquatic Geomicrobiology, Friedrich Schiller University, Jena, 07743, Germany
[2]Department of Hydrogeology, Institute of Geosciences, Friedrich Schiller University, Jena, 07743, Germany
[3]The German Centre for Integrative Biodiversity Research (iDiv) Halle-Jena-Leipzig, Leipzig, 07743, Germany

*Correspondence to*: Kirsten Küsel (kirsten.kuesel@uni-jena.de)

**Abstract.** The formation of Fe(III) oxides in natural environments occurs in the presence of natural organic matter (OM), resulting in the formation of OM-mineral complexes that form through adsorption or coprecipitation processes. Thus,
microbial Fe(III) reduction in natural environments most often occurs in the presence of OM-mineral complexes rather than pure Fe(III) minerals. In this study we investigated to which extent the content of adsorbed or coprecipitated OM on ferrihydrite influences the rate of Fe(III) reduction by *Shewanella oneidensis* MR-1, a model Fe(III)-reducing microorganism, in comparison to a microbial consortium extracted from the acidic, Fe-rich Schlöppnerbrunnen fen. We found that increased OM contents led to increased rates of microbial Fe(III) reduction by *S. oneidensis* MR-1 in contrast to earlier findings with the
model organism *Geobacter bremensis*. Ferrihydrite-OM coprecipitates were reduced slightly faster than ferrihydrites with adsorbed OM. Surprisingly, the complex microbial consortia stimulated by a mixture of electrons donors (lactate, acetate, and glucose) mimics *S. oneidensis* under the same experimental Fe(III)-reducing conditions suggesting similar mechanisms of electron transfer whether or not the OM is adsorbed or coprecipitated to the mineral surfaces. We also followed potential shifts of the microbial community during the incubation via 16S rRNA gene sequence analyses to determine variations due to
the presence of adsorbed or coprecipitated OM-ferrihydrite complexes in contrast to pure ferrihydrite. Community profile analyses showed no enrichment of typical model Fe(III)-reducing bacteria, such as *Shewanella* sp. or *Geobacter* sp., but an enrichment of fermenters (i.e. *Enterobacteria*) during pure ferrihydrite incubations which are known to use Fe(III) as an electron sink. Instead, OM-mineral complexes favoured the enrichment of microbes including *Desulfobacteria* and *Pelosinus* sp., both of which can utilize lactate and acetate as an electron donor under Fe(III) reducing conditions. In summary, this
study shows that increasing concentrations of OM in OM-mineral complexes determines microbial Fe(III) reduction rates and shapes the microbial community structure involved in the reductive dissolution of ferrihydrite. Similarities observed between the complex Fe(III)-reducing microbial consortia and the model Fe(III)-reducer *S. oneidensis* MR-1 suggest electron shuttling mechanisms dominate in OM-rich environments, including soils, sediments, and fens, where natural OM interacts with Fe(III) oxides during mineral formation.

## 1 Introduction

Iron is the fourth most abundant element in the Earth's crust, indicating the role of Fe-cycling in nature is both ubiquitous and widespread. The majority of which are dispersed as iron minerals in natural environments, including soils and sediments with ferrous iron [Fe(II)] and ferric iron [Fe(III)] as the primary redox states (Davison, 1993; Stumm and Sulzberger, 1992; Weber
et al., 2006a). Microbially mediated and abiotic redox cycling of Fe play crucial roles in global biogeochemistry, for example, preservation and degradation of organic matter, and the fate of nutrients as well as contaminants for living organisms (Borch et al., 2010; Melton et al., 2014). Iron minerals such as Fe(III) oxides exhibit a high reactivity towards dissolved organic




matter (OM), thus Fe(III) oxides are either partially or completely covered by OM in natural environments (Eusterhues et al., 2005; Kaiser and Zech, 2000; Lalonde et al., 2012; Torn et al., 1997). Fe(III) oxides enveloped by OM potentially lead to

changes in the surface properties in comparison to non-modified Fe(III) oxides, which ultimately may influence mobility, solubility, and aggregation (Eusterhues et al., 2014; Narvekar et al., n.d.; Pédrot et al., 2011; Vindedahl et al., 2016). The poorly crystalline ferrihydrite is one of the most common Fe(III) oxides and typically forms aggregates of nanometer-sized individual crystals (Bigham et al., 2002; Cornell and Schwertmann, 2003; Jambor and Dutrizac, 1998). The coprecipitation of OM results in the adsorption and occlusion of organic molecules within the interstices between individual ferrihydrite

crystals, while adsorption of OM occurs on pre-existing ferrihydrite surfaces (Eusterhues et al., 2014). Moreover, the presence of dissolved OM impedes ferrihydrite growth (Cismasu et al., 2011; Eusterhues et al., 2008; Mikutta et al., 2008; Schwertmann et al., 2005), therefore, ferrihydrite with coprecipitated OM have smaller crystal sizes and more crystallographic defects. Since ferrihydrite is formed usually in OM-rich environments coprecipitation is the common process in nature (Eusterhues et al., 2008). Many properties differ between coprecipitated ferrihydrite compared to pure ferrihydrite, thus the accessibility and

solubility of ferrihydrite surfaces along with the accessibility of the adsorbed/occluded OM to microbes, redox-active shuttling compounds, extracellular enzymes, or reducing agents potentially differ from ferrihydrites with purely adsorbed OM.

Not only do the mechanisms through which OM interacts with Fe(III) oxides, e.g. adsorbed or coprecipitated, influence the reactivity of the mineral-OM complex, the composition of OM may also effect its reactivity. The redox reactivity of OM is attributed mostly to quinones. Past experiments have often been performed using humic acids, i.e., alkaline extracts

of natural organic material, which usually contain a higher concentration of the aforementioned redox active functional groups, than natural OM from soil (Fritzsche et al., n.d.; Klüpfel et al., 2014; Lehmann and Kleber, 2015; Piepenbrock et al., 2014). Humic acids should therefore be more effective in electron shuttling than natural soil organic matter. Previous studies have shown that the presence of dissolved humic acids leads to complexation of Fe(II) (Royer et al., 2002), and complexation and dissolution of Fe(III) (Jones et al., 2009), but also potentially enhance Fe(III) reduction via electron shuttling (Hansel et al.,

2004; Jiang and Kappler, 2008; Lovley et al., 1996; Roden et al., 2010). Furthermore, the concentrations of dissolved humic acid or the mineral/humic acid ratios where shown to influence the possibility of humic acids increasing Fe(III) reduction rates or not (Amstaetter et al., 2012; Jiang and Kappler, 2008). Despite the aforementioned studies, the influence of mineral bound OM on Fe(III) reduction rates and mineral transformation is not fully understood in regards to the broad diversity of Fe(III) reducing microorganisms present in nature (Weber et al., 2006a, 2006b).

Fe(III) oxides as well as OM-mineral complexes are not able to pass through the outer membrane of Fe(III)-reducing microorganisms, therefore different electron transfer pathways have evolved to alleviate this physiological problem (Poggenburg et al., 2016; Weber et al., 2006b). Several mechanisms used as electron transfer pathways have been characterized including (i) direct transfer of electrons via enzymes anchored to the outer membrane of the bacterial cell (Myers and Nealson, 1988; Nevin and Lovley, 2000) or conductive nanowires (El-Naggar et al., 2010; Gorby et al., 2006; Reguera et

al., 2005) and (ii) mediated electron transfer via exogenous electron shuttling compounds found in natural OM (Lovley et al., 1996), endogenous electron shuttling compounds, for example flavins (von Canstein et al., 2008; Kotloski and Gralnick, 2013;



Marsili et al., 2008; Nevin and Lovley, 2002; Newman and Kolter, 2000), or the use of Fe(III)-chelating compounds, such as siderophores (Kraemer, 2004; Lovley et al., 1994; Nevin and Lovley, 2002).

75       *Shewanella* and *Geobacter* species represent two of the best studied model Fe(III) reducing microorganisms. *Shewanella* species have been shown to utilize endogenous electrons shuttles, such as flavins, rather than transferring electron via direct contact between the mineral and the microbe (von Canstein et al., 2008; Kotloski and Gralnick, 2013). Not surprisingly, microbial reduction of Fe(III) oxides like nanometer-sized lepidocrocite and ferrihydrite by *Shewanella* species can be boosted by the presence of humic acids (Amstaetter et al., 2012; Pédrot et al., 2011; Shimizu et al., 2013) and quinone moieties (Newman and Kolter, 2000) when the amount of dissolved organic carbon is greater than a threshold concentration

of 5-10 mg C L$^{-1}$ (Jiang and Kappler, 2008). Conversely, a study by Eusterhues, et al (2014) reported high C/Fe ratios in coprecipitated and adsorbed ferrihydrite complexes inhibited the reduction of ferrihydrite by *G. bremensis*. Poggenburg et al. (2016) produced ferrihydrite OM complexes with different organic materials (extracted extracellular polymeric substances, water extracts of soil litter) and used *S. putrefaciens* and *G. metallireducens* to reduce them. Their results confirmed that ferrihydrite-bound OM increases reduction rates of *Shewanella*, whereas it protects the mineral surface from reduction by

*Geobacter*. For reduction by *Shewanella* this study found that reduction rates were correlated to the aromaticity of the mineral-bound OM. *Geobacter* species have been shown to use direct contact and conductive nanowire structures for electron transfer (Boesen and Nielsen, 2013; Malvankar et al., 2011). Unlike *Shewanella, Geobacter* do not produce endogenous electron shuttling or chelating compounds (Nevin and Lovley, 2000). *Geobacter* are capable of reducing both humic acid (Jiang and Kappler, 2008; Lovley et al., 1996) and extracellular quinone moieties (Scott et al., 1998) in the absence of Fe(III) oxides. The

decreased rates of Fe(III) reduction of OM-ferrihydrite complexes by *G. bremensis* compared to pure ferrihydrite minerals suggest that *Geobacter* do not use external electron shuttles in natural environments rich in OM and Fe(III) oxides. Since the effects of OM on Fe(III) reduction rates vary between the two model organisms presumably due to the various mechanisms of electron transfer used, we should move forward by studying the response of a diverse community of Fe(III) reducers living in a habitat rich in Fe(III) oxides and OM. Taken together, these studies indicate (i) that it depends on the microorganisms present

and their Fe respiratory pathways, whether OM on Fe(III) oxide mineral surfaces hinders or promotes Fe(III) oxide reduction. (ii) When electron shuttling is the dominant electron transfer mechanism, a high content of redox active groups in the mineral-associated OM speeds up reduction.

      In this study, we used the same experimental design described in the earlier study of Eusterhues et al., 2014, but replaced *G. bremensis* with *S. oneidensis* MR-1 to test the effect of increasing amounts of adsorbed and co-precipitated OM

on microbial Fe(III) reduction. In a parallel set of incubation studies we used a microbial consortia as inoculum derived from an iron-rich peatland where the microbial reduction of Fe(III) is the dominant electron accepting processes for the degradation of OM under anoxic conditions. This Schlöppnerbrunnen fen has been studied in great detail during the last 10 years, however, the impact of OM on the microbially-mediated Fe(III) reduction processes in this fen are not fully understood. We compared rates of Fe(III) reduction of ferrihydrite adsorbed and co-precipitated with OM with pure ferrihydrite and followed qualitative

and quantitative changes in the microbial community during incubation. To our great surprise, rates observed with *Shewanella*




matched nearly exactly those observed with the complex consortia derived from Schlöppnerbrunnen fen, although *Shewanella* was not the dominant Fe(III)-reducing microorganism detected at the beginning and end of incubation.

## 2 Methods

### 2.1 Study site and sampling procedure

Peat cores were obtained from the Schlöppnerbrunnen fen (Northern Bavaria, Germany; 50°07'55''N, 11°52'52''E), a minerotrophic, slightly acidic (pH ~5) described in detail (Blodau et al., 2004; Eusterhues et al., 2014; Hausmann et al., 2016; Küsel et al., 2008; Loy et al., 2004; Pester et al., 2012), using a Pürkheimer soil corer in May 2016. Briefly, granite bedrock is covered by a fibric histosol, often referred to as peat, with a thickness between 40 cm and 70 cm. This fen is fed by anoxic, Fe(II)-rich groundwater (Küsel et al., 2008). The sampling site within the fen used in the current study (site "M") is positioned

southwest of previously described drying-rewetting experimental plots (Knorr and Blodau, 2009; Reiche et al., 2008). Previous studies have reported that high concentrations of Fe(II) (up to 7.4 mM, mean 2001-2004 0.3 mM), DOC (up to 48 mM, mean 2001-2004 5.1 mM) and $C_{org}$ (mean 37%) are routinely found at site "M" (Küsel et al., 2008; Reiche et al., 2008). Peat cores were transported on ice under anoxic conditions ($N_2$) and peat samples were either processed or stored at -20°C until use.

### 2.2 Microbial extraction from peat core samples

The peat microbial community was extracted using a standard protocol adapted to soil microbial extractions.  Briefly, a 20 mL syringe, with tip removed, was used as a manual short core sampler to remove 20 mL fractions (~25 g) from bulk peat cores and added to sterile, anoxic 125 mL serum bottles containing 100 mL 0.85% NaCl solution and 8 g sterile glass beads.  The slurry was flushed with Ar and shaken at 225 rpm at 4 °C overnight.   Samples were transferred to 50 mL Falcon tubes and centrifuged at increasing speeds to observe sedimentation rates.  The samples were initially centrifuged at 900 g for 5 minutes.

The sample was decanted into a fresh Falcon tube and the supernatant was centrifuged again at 1200 g for 10 min.  Next, the solid phase was resuspended in 10 mL anoxic 0.85% NaCl solution.  The extract was transferred to 30 mL serum bottles and flushed with Ar to ensure anoxic conditions (Supplementary Fig. 1).  The presence of bacteria in the peat microbial extract was confirmed via SYTO (50 μM) staining and microscopy.

        The peat microbial extractions were performed in duplicated triplicates to ensure sufficient inoculum was available

for subsequent incubations.  Genomic DNA was obtained from the peat microbial extract using the PowerSoil DNA isolation kit (MO BIO Laboratories, Carlsbad, CA, USA) according to manufacturer's instructions and stored at -20°C.  DNA extracts were subsequently used as a template for quantitative PCR (qPCR) reactions targeting bacterial 16S rRNA genes to quantify abundance of bacteria over time or microbial community analysis via Illumina sequencing.



**2.3 Synthesis of ferrihydrite and organic matter (OM)-mineral complexes**

A 2-line ferrihydrite was produced as previously described by Eusterhues, et al. (2014). Briefly, the 2-line ferrihydrite was produced via titration of 0.01 M $Fe(NO_3)_3$ solution with 0.1 M NaOH to pH 5 with constant stirring. Ferrihydrites with varying concentrations of adsorbed OM were produced by mixing forest floor extract solutions containing increasing C concentrations with suspensions of freshly precipitated 2-line ferrihydrite at pH 5. Molar C/Fe ratios of these initial solutions were AFhA 0.4, AFhB 1.3, and AFhD 4.2. Coprecipitated ferrihydrites were produced by dissolving $Fe(NO_3)_3$ in forest floor extract solutions

of increasing concentrations and adding 0.1 M NaOH under vigorous stirring until the solution reached pH 5. Molar C/Fe ratios of these initial solutions were CFhA 0.4, CFhB 1.3, and CFhD 4.2. The solid/solution (g/L) ratio for all synthesized minerals ranged between 0.3 and 1.6. The solid products were separated using centrifugation, washed two times with deionized $H_2O$, freeze dried, and stored in hungate tubes under anoxic conditions ($N_2$). The C content was measured using a CN-Analyzer (Vario EL, Elementar-Analysensysteme, Hanau, Germany) and the Fe content by ICP-OES after total digestion (Spectroflame,

Spectro, Kleve, Germany). A characterization of ferrihydrite-OM complexes by FTIR and XPS can be found in Eusterhues et al. (2014). Note, the extraction of soil organic matter was also performed as described by Eusterhues et al. (2014) using the Oa and Oe layers of a Podzol under spruce located near Freising, Germany.

**2.4 Microbial OM ferrihydrite reduction experiments**

We used *S. oneidensis* MR-1 as model Fe(III)-reducer since it is well-characterized and able to utilize a variety of carbon

substrates as electron donors (Myers and Nealson, 1988, 1990; Venkateswaran et al., 1999). Liquid aerobic pre-cultures were grown in 50 mL of lysogeny broth (Luria Bertani medium; LB medium) containing 10 g tryptone, 10 g NaCl, 5 g yeast extract. These cultures was incubated on a rotary shaker at 150 rpm and the cell concentration was determined by optical density (OD) measurements at 600 nm. For the Fe(III) reduction experiments 2 ml of LB-grown pre-culture was harvested after 14 h at late exponential growth phase and centrifuged at 10,000 g for 5 min. Cells were washed twice with defined freshwater medium,

resuspended in 2 ml of medium and diluted to a final concentration of 2 x $10^5$ cells $ml^{-1}$ in culture tubes containing defined freshwater medium. Defined freshwater medium (pH 6.8; *Geobacter* medium ATCC 1957) was prepared with a combination of electron donors (2 mM lactate, 2mM acetate, 2 mM glucose) and $NaHCO_3$ to adjust the pH.

     In the current study the microbial community extracted from peat cores collected in the Schlöppnerbrunnen fen as well as *S. oneidensis* MR-1 were used as the inoculum in incubations containing either ferrihydrite, OM-adsorbed ferrihydrite,

or OM-co-precipitated ferrihydrite. For OM-adsorbed ferrihydrite incubations, three different concentrations of organic matter were used: 44 mg/g, 105 mg/g. and 181 mg/g TOC. Similarly, three different concentrations of organic matter were used in OM-co-precipitated ferrihydrite incubations: 44 mg/g, 98 mg/g. and 182 mg/g TOC. For all incubations a defined freshwater medium (pH 6.8; *Geobacter* medium ATCC 1957) containing 1.5 g $L^{-1}$ $NH_4Cl$ and 0.1 g $L^{-1}$ KCl, was used. After autoclaving and cooling under anoxic conditions ($N_2$ /$CO_2$ (80 / 20 v/v) atmosphere), 30 mL $L^{-1}$ of 1 M $NaHCO_3$ (autoclaved, $CO_2$), 10

mL $L^{-1}$ Wolfe's vitamin solution (ATCC 1957), and 10 mL $L^{-1}$ modified Wolfe's minerals (ATCC 1957). Filter-sterilized



(0.2 µm) sodium-acetate (2 mM), sodium lactate (2mM), and glucose (2mM) were added as carbon sources to all incubations to ensure standardization. $NaH_2PO_4$ was not added to circumvent interaction of $PO_4^{3-}$ with ferrihydrite.

10 mL medium was added to pre-sterilized (6 h, 180 °C) 21 mL hungate tubes containing pre-weighed ferrihydrite, OM-adsorbed ferrihydrite (TOC: 44 mg/g C, 105 mg/g C, 181 mg/g C), or OM-co-precipitated ferrihydrite (TOC: 44 mg/g C,
98 mg/g C, 182 mg/g C) with constant flushing ($N_2$). The concentration of Fe per tube ranged from 0.63-0.85 mmol. The hungate tubes were sealed with butyl rubber stoppers and flushed with sterile $N_2/CO_2$ (80%/20% *v/v*) for ~2 h to ensure anoxic conditions. Either peat microbial extract (5% *v/v*) or *S. oneidensis* MR-1 were used as inoculum in separate incubations containing ferrihydrite, OM-adsorbed ferrihydrite (TOC: 44 mg/g, 98 mg/g, 182 mg/g), or OM-co-precipitated ferrihydrite (TOC: 44 mg/g, 105 mg/g, 181 mg/g). All treatments were performed in triplicate and incubated horizontally at room
temperature in the dark and shaken periodically. Fe(III) reduction was monitored approximately every 24 hours using the phenanthroline assay (Tamura et al., 1974) (data not shown). Fe(II) formation kinetics were used as analogues for Fe(III) reduction. Apparent initial reaction rates were estimated by fitting linear regression lines to Fe(II) [mM]) versus time [hrs] for all data points. The slope of the line represents the initial reaction rate. At the end of incubation (~12 days), solid remnants were freeze dried and stored under $N_2$ until X-ray diffraction (XRD) measurements were performed (see description below).

**2.5 DNA extraction**

Genomic DNA was extracted from three biological replicates at 3 different time points ($T_0$, $T_{middle} - 136$ h, $T_{end} - 296$ h) from all incubations inoculated with either peat microbial extract or *S. oneidensis* MR-1 using the PowerSoil DNA isolation kit (MoBio; Catalog # 12888-50). The genomic DNA was stored at -20°C.

**2.6 Quantitative PCR**

Bacterial16S rRNA gene copy and transcript numbers of the total peat microbial community were determined by quantitative PCR using a Mx3000P instrument (Agilent) with Maxima SYBR Green Master Mix (Fermentas). Bacterial 16S rRNA genes were amplified using primer pairs Bac8Fmod/Bac338Rabc (Daims et al., 1999; Loy et al., 2002) according to cycling conditions and standards previously described (Herrmann et al., 2012). *S. oneidensis* MR-1 16S rRNA gene copy numbers were determined with primer pairs She12F/She220R (Himmelheber et al., 2009). 2 to 20 ng of genomic DNA was used as
template. Standard curves were produced using serial dilutions of representative plasmid mixtures and were linear for both primer sets from $5x10^8$ to $5x10^2$ copies with $R^2$ values of 0.999-1.000 and the qPCR performed with efficiencies between 80-90%. To monitor temporal changes in transcriptional activity, transcript numbers were normalized to copy numbers of the corresponding gene copy number per genome: 4.02 for bacterial 16S rRNA and 9 for *S. oneidensis* 16S rRNA gene (https://rrndb.umms.med.umich.edu, as retrieved on June 13th 2017). qPCR was performed on subsamples from $T_0$, $T_{middle}$
(136 h), and $T_{end}$ (296 hr). One-way analysis of variance (ANOVA) was used to compare treatments using the calculated quantification results and determine statistical significance ($p<0.05$).



## 2.7 Illumina MiSeq Amplicon sequencing

16S rRNA gene-targeting amplicon sequencing was done using a bacteria-specific primer pair (341f [S-D-Bact-0341-b-S]/805r [S-D-Bact-0785-a-A]) (Klindworth et al., 2013). Amplicon sequencing was performed by LGC Genomics GmbH (Berlin,

Germany). PCR products were generated starting from about 5 ng of extracted genomic DNA extract, 15 pmol of forward and reverse primer in 20 uL of 1 × MyTaq buffer containing 1.5 units MyTaq DNA polymerase (Bioline, City, Country) and 2 µl of BioStabII PCR Enhancer (Sigma-Aldrich, Munich, Germany). For each sample, the forward and reverse primers had the same 10-nt barcode sequence. PCRs were carried out for 30 cycles using the following parameters: 2 min 96°C pre-denaturation; 96°C for 15 s, 50°C for 30 s, 70°C for 90 s. DNA concentrations of amplicons of interest were quantified semi-

quantitatively via gel electrophoresis. ~20 ng amplicon DNA of each sample were pooled for up to 48 samples carrying different barcodes. If needed PCRs showing low yields were further amplified for an additional 5 cycles. The amplicon pools were purified with one volume AMPure XP beads (Beckmann-Coulter, Krefeld, Germany) to remove primer dimer and other small mispriming products, followed by an additional purification on MinElute columns (Qiagen, Dusseldorf, Germany). About 100 ng of each purified amplicon pool DNA was used to construct sequencing libraries using the Ovation Rapid DR

Multiplex System 1–96 (NuGEN, Leek, Netherlands). Illumina libraries were pooled and size selected by preparative gel electrophoresis. Sequencing was done on an Illumina MiSeq using V3 chemistry (Illumina) in paired-end mode (2 x 300 bp). Amplicon sequencing was performed genomic DNA samples taken at $T_0$ and $T_{end}$ (296 hr).

## 2.8 Microbiome profiling

Sequence characteristics were checked using *prinseq* (v. 0.20.4) (Schmieder and Edwards, 2011). Adaptor sequences still

present were subsequently removed with *cutadapt* (v. 1.7) (Martin, 2011). Paired-end assembly and subsequent quality control were done using *vsearch* (v. 1.10.2) (Rognes et al., 2016) using the implemented -fastq_mergepairs (-fastq_minmergelen = expected amplicon size) and -fastq_filter (-fastq_maxee = 0.5) functions. Chimera removal was done with *uchime* (v. 4.2.40) (Edgar et al., 2011). OTU clustering with a defined threshold was omitted, instead sequences were dereplicated to streamline the dataset. Deduced dereplicated sequences were treated as being representative for sequence populations featuring an identity

of 100% (in the following referred to as OTU). The dataset was stripped by removing OTUs that were only identified in single samples. Sequences from this simplified dataset were taxonomically assigned against *SILVA* (release 123, as of September 2016) using the *uclust* algorithm (Edgar, 2010) implemented in *vsearch* applying a minimum similarity of 90%. An OTU table including determined taxonomic affiliations and available metadata was assembled for downstream analysis in *qiime* (v. 1.9.2) (Caporaso et al., 2010) and *phyloseq* (v. 1.10) (McMurdie et al., 2013). Statistical analyses were carried out using the R

statistical software framework (R core team, 208) and the following packages, including respective dependencies, *phyloseq*, *vegan* (v. 2.4-3) (Oksanen et al., 2017) , and *DESeq2* (v. 1.10.1) (Love et al., 2014). Differential abundances of taxonomic



groups against defined parameters (ferrihydrite complexation, C/Fe ratio) were done using negative binomial distributions combined with Wald significance tests, and parametric fitting for determined dispersions.

**2.9 Sequence data deposition**

Generated amplicon sequencing data was deposited as raw data at NCBI SRA and are available under the following BioProject accession number: PRJNA391563

**2.10 X-ray diffraction (XRD) measurements**

X-ray diffraction measurements were performed with a D8 Advance DaVinci diffractometer (Bruker AXS, Karlsruhe, Germany) using Cu-Kα radiation at 40 kV and 40 mA, scanning from 5 to 80 °2Θ with increments of 0.02 °2Θ. Mineral
identification was based on the PDF database of 2011.

**3 Results**

**3.1 Microbial Fe(III) reduction by *Shewanella oneidensis* MR-1 and by the microbial consortia extracted from peat**

We compared the Fe(III) reduction capacity of the model organism *S. oneidensis* MR-1 with a peat microbial consortia extracted from an anoxic iron-rich peatland for incubations containing pure ferrihydrite (Fh control), coprecipitated OM-
ferrihydrite complexes (CFhA (44 mg/g C), CFhB (98 mg/g C), CFhD (182 mg/g C)), or adsorbed OM-ferrihydrite complexes (AFhA (44 mg/g C), AFhB (105 mg/g C), AFhD (181 mg/g C)). The Fe(III) reduction patterns of *S. oneidensis* MR-1 and the peat microbial consortia were nearly identical. The lowest Fe(III) reduction rate was observed in *S. oneidensis* MR-1 incubations with pure ferrihydrite with a reduction rate of 0.021 mmol hr$^{-1}$ compared to 0.031 mmol hr$^{-1}$ for incubations with our peat microbial consortia (Fig. 1, Fig. 2, Table 1). The highest Fe(III) reduction rates were observed in incubations
containing coprecipitated OM-ferrihydrite complexes, with increasing rates of reduction corresponding to increased C/Fe ratios (from 44 to 182 mg g C). Moreover, the highest rates of Fe(III) reduction were observed in incubations containing coprecipitated OM-ferrihydrite (182 mg/g C) and either the peat microbial consortia or *S. oneidensis* MR-1 (0.055 mmol hr$^{-1}$ and 0.050 mmol hr$^{-1}$, respectively) (Fig. 1a, Fig. 2a, Table 1). Similar patterns occurred in incubations with adsorbed OM-ferrihydrite complexes, such that the higher the C/Fe ratios the higher the rate of Fe(III) reduction (from 44 to 181 mg g C).
The rate of Fe(III) reduction was slightly higher in incubations with peat microbial extract compared to *S. oneidensis* MR-1 (0.053 mmol hr$^{-1}$ and 0.043 mmol hr$^{-1}$, respectively) (Fig. 1b, Fig. 2b,Table 1). Surprisingly, the rates of Fe(III) reduction in incubations with the peat microbial consortia and *S. oneidensis* MR-1 increased with increasing TOC concentrations (and C/Fe ratios) regardless of whether adsorbed OM-ferrihydrite complexes or coprecipitated OM-ferrihydrite complexes were used as the terminal electron acceptor(Fig. 1, Fig. 2, Table 1).



## 3.2 Quantitative changes of 16S rRNA genes in incubation experiments

We followed the abundance of bacterial 16S rRNA gene copies over time in incubations inoculated with either *S. oneidensis* MR-1 or the microbial consortia extracted from peat to determine potential effects of coprecipitated or adsorbed OM-ferrihydrite complexes on microbial abundances. In general, bacterial 16S rRNA copy numbers increased over time in all incubations with noticeable increases in gene abundances at the end of incubation and slight variations dependent on the TOC concentration of the OM coprecipitated or adsorbed to the surface of ferrihydrite (Fig. 3, Fig. 4). In incubations with pure ferrihydrite, *S. oneidensis* MR-1 16S rRNA gene copy numbers increased four orders of magnitude from $7.90 \times 10^7$ cells mL$^{-1}$ at $T_0$ to $1.99 \times 10^{11}$ cells mL$^{-1}$ at $T_{end}$ (296 h). *S. oneidensis* MR-1 16S rRNA gene copy numbers increased approximately five orders of magnitude in incubations with coprecipitated or adsorbed OM-ferrihydrite complexes with the highest C/Fe ratios ($8.10 \times 10^7$ cells mL$^{-1}$ at $T_0$ to $8.47 \times 10^{12}$ cells mL$^{-1}$ at $T_{end}$ and $8.10 \times 10^7$ cells mL$^{-1}$ at $T_0$ to $1.24 \times 10^{12}$ cells mL$^{-1}$ at $T_{end}$, respectively). In incubations with pure ferrihydrite and the peat microbial consortia, bacterial 16S rRNA gene copy numbers only increased two orders of magnitude from $6.23 \times 10^{10}$ cells mL$^{-1}$ at $T_0$ to $2.02 \times 10^{12}$ cells mL$^{-1}$ at $T_{end}$. Bacterial 16S rRNA gene copy numbers increased roughly three orders of magnitude in incubations using the peat microbial consortia as inoculum and adsorbed OM-ferrihydrite complexes with the highest C/Fe ratios ($6.23 \times 10^{10}$ cells mL$^{-1}$ at $T_0$ to $2.58 \times 10^{13}$ cells mL$^{-1}$ at $T_{end}$), however, the gene copy numbers increased approximately four orders of magnitude in incubations with coprecipitated OM-ferrihydrite with the highest C/Fe ratios ($6.23 \times 10^{10}$ cells mL$^{-1}$ at $T_0$ to $1.36 \times 10^{14}$ cells mL$^{-1}$ at $T_{end}$, respectively). Thus, increases in gene copy numbers correlate to the increased Fe(III) reduction rates observed across all incubations with OM-ferrihydrite complexes and pure ferrihydrite.

## 3.3 Microbial community profiling of the microbial consortia in incubation experiments

Incubations with absorbed and coprecipitated OM-ferrihydrite and varying C/Fe ratios led to distinct shifts in microbial community structure when compared to pure ferrihydrite incubations and the initial microbial consortia extracted from peat (Fig. 5). Alpha diversity indices addressed intra-sample diversity either based on the observed number of taxa according to the applied OTU definition (observed OTUs) or based on extrapolations of observed taxa numbers putting special emphasis on rare taxa (Chao1, ACE). Comparing the different incubations with the initial peat microbial extract (Observed OTUs: 52064, Chao1: 801599, ACE: 819056, except AFhB) revealed decreased diversities in all cases (Observed OTUs: 23120-46770, Chao1: 334438-545083, ACE: 266190-455099) except AFhB (Fig. 5a). Intra-sample diversity measures were complemented by assessing the evenness of datasets by determining the entropy of respective datasets (Shannon) and determining the probability of not randomly subsampling sequences of same taxonomic origin when two sequences are sampled (Simpson). Respective values matched observed intra-sample diversities. Calculated Shannon and Simpson indices showed strongly increased evenness for the initial peat microbial extract (Shannon: 9.47, Simpson: 0.991) against all incubations (Shannon: 6.77-7.48, Simpson: 0.88-0.966)f, excluding AFhB. Beta diversity analysis is a common approach to gain insights about inter-sample diversity when dealing with complex datasets such as microbial community profiles (Fig. 5b). Corresponding analyses



based on Jensen-Shannon divergence highlighted a minor clustering of microbial communities linked to coprecipitated OM-ferrihydrite incubations in case of low and medium C/Fe ratios (CFhA, CFhB). Community profiles originating from incubations supplemented with OM-ferrihydrite having a high C/Fe ratio revealed a clustering independent of ferrihydrite

complexation (AFhD, CFhD). The observation of community profiles from incubations clustered distinctly from communities present in the initial peat microbial extract matched the abovementioned differences with respect to intra-sample diversity and evenness (Fig. 5a).

Looking at underlying community profiles from a taxonomic perspective revealed a dominance of few phyla when considering determined community profiles as a whole (Fig. 6A). Dominant phyla included especially Proteobacteria (57.9-

95.4% relative abundance), Firmicutes (3.1-41.6%), and to a lesser extent Bacteroidetes (up to 0.6% [AFhD, CFhD]) and Acidobacteria (4.5% [peat microbial extract]). Looking at the taxonomic composition at a more resolved level (family-level) showed that carried out incubations led in all cases to a strong increase of Pseudomonadaceae in relative abundance within the Gamma-Proteobacteria (e.g. AFhA 58.1%, in comparison to peat microbial extract 5.3%) (Fig. 6b). Pronounced increases were also seen for Veillonellaceae (AFhA, CFhA, CFhB [24.6-35.5%], in comparison to peat microbial extract [< 0.00001%], and

Fh [< 0.00001%]), Bacillaceae (AFhA, AFhB, CFhA [5.4-12.0%], in comparison to peat microbial extract [< 0.00001%], and Fh [3.6%]), and Desulfovibrionaceae (CFhA, CfhB [1.8-2.5%], in comparison to peat microbial extract [< 0.001%], and Fh [< 0.00001%]). Increased abundances of Veillonellaceae were correlating with low and medium C/Fe ratios, while incubations with high C/Fe ratios favoured elevated relative abundances of Enterobacteriaceae, when compared with low and medium C/Fe ratios. Groups that appeared to be enriched in the incubation with pure ferrihydrite, for example Burkholderiaceae, were

found to show only minor abundances in any of the other setups.

Considering the results from beta diversity analysis and community profiling, we were interested in assessing the differential abundance of genus-level taxonomic groups to check for the potential enrichment of taxonomic groups that are linked to iron cycling when comparing the pure ferrihydrite incubation and OM-ferrihydrite incubations. Differential abundances were examined by negative binomial distributions (Fig 6c). It was apparent that numerous genus-level taxonomic

groups were underrepresented when CFh and AFh are contrasted against Fh. This was consistent with reduced intra-sample diversities for adsorbed and co-precipitated incubations in comparison to the pure ferrihydrite incubation (Fig. 5a). Most underrepresented genus-level taxonomic groups were linked to Proteobacteria and especially Enterobacteriales/Enterobacteriaceae including: *Ewingella, Rahnella, Serratia, and Yersinia.* This pattern was similar for CFh against Fh, as well as AFh against Fh. In most cases log2 fold changes ranged between -5 and -10 for underrepresented groups.

The number of overrepresented taxonomic groups was in general lower than the number of underrepresented groups. This finding was much more pronounced for AFh vs Fh than CFh vs Fh. Only three overrepresented genus-level taxonomic groups were identified for AFh vs Fh, namely *Pelosinus*, *Pseudomonas*, and *Telmatospirillum* with log2 fold changes above 5. The same groups were also identified to be enriched in case of CFh vs Fh. However, overall the number of enriched taxonomic groups was higher including various Firmicutes such as *Bacillus*, *Clostridium*, *Lachnoclostridium,* and *Paenibacillus*. Few

genus-level groups comprised over- and underrepresented OTUs. This was the case for *Bacillus*, *Paenibacillus*, and





*Pseudomonas*. In the case of *Pseudomonas,* one particular OTU was found to be enriched. Consequently, this indicates that pronounced increases in the abundance of *Pseudomonaceae* in adsorbed and co-precipitated OM-ferrihydrite are apparently mediated by a well-defined sub population of *Pseudomonas*.

### 3.4 XRD measurements

X-ray diffraction patterns of the solid remnants after reduction showed two very broad peaks of ferrihydrite as well as overlying sharp peaks of different salts (halite, nahcolite, salammoniac), which derive from the medium. Goethite, other Fe(III) oxides or Fe(II) minerals were not found.

### 4 Discussion

Dissolved, particulate and mineral-associated OM plays a crucial role in many biogeochemical processes in soils and
sediments. The ability of OM to serve as a terminal electron acceptor for microbial respiration in anoxic environments is of special importance and has been well characterized (Lovley et al., 1996). However, most experiments utilizing OM as a terminal electron acceptor have been performed with alkaline extracts: Reduced humic acids have been shown to transfer electrons abiotically to terminal electron acceptors, such as metals, thereby functioning as electron shuttles between microbes and electron acceptors (Kappler et al., 2004). Humic acids, therefore, possess a redox-mediating ability which positively
affects the cycling of redox-active minerals, including Fe(III) and Mn(IV)/Mn(III), in conjunction with transformation of organic and inorganic compounds. Subsequent studies have shown that reduced forms of humic substances can serve as electron donors for anaerobic organisms growing on a variety of terminal electron acceptors, including nitrate and fumarate (Coates et al., 2002). Under these conditions, microorganisms utilize the reduced humic substances as an energy source, while using a readily degradable carbon source, such as acetate, for growth. This unique ability to use humic substances as an electron
donor provides a possible competitive advantage over other heterotrophs in the environment which are only capable of utilizing a single organic compound as both the carbon and energy source. Previous studies have revealed that some Fe(III)-reducing microorganisms, such as *S. alga, G. metallireducens,* and *Geothrix fermentans*, have the ability to undergo dissimilatory humic reduction as well as the ability to couple reduced humic oxidation with nitrate reduction. Moreover, *Paracoccus denitrificans*, a known denitrifier, is unable to utilize humics or Fe(III) as a terminal electrons acceptors, however, these microorganisms are
capable of using reduced humics as an electron donor for denitrification (Coates et al., 2002; Lovley et al., 1999).

The current study provides clear evidence that the microbial community found in the Fe-rich Schlöppnerbrunnen fen not only has the potential to reduce Fe(III)-oxides complexed with organic matter to the same extent as the model Fe(III)-reducer, *S. oneidensis*, but the microbes also increase in abundance under the experimental growth conditions used. We used a mixture of electron donors (glucose, lactate, and acetate) to stimulate a great variety of peat microorganisms capable of
dissimilatory Fe(III) reduction including those that couple Fe(III) reduction with growth, but also fermenting microorganisms and others that just use Fe(III) oxides as electron sink (Hori et al., 2015; Lentini et al., 2012). Earlier incubation studies with peat soil from the Schlöppnerbrunnen fen demonstrated that amendment with all these substrates stimulate Fe(III) reduction



with glucose showing the most rapid increase of Fe(II) production (Küsel et al., 2008). Glucose was completely consumed within 3 days of incubation, yielding acetate and $CO_2$ as end products, and $H_2$ and ethanol as transient products. Supplemental

lactate yielded acetate as a main product (Küsel et al., 2008). Both acetate and $H_2$ are preferred substrates for many Fe(III)-reducing microorganisms including *Shewanella* and *Geobacter* species. Thus, the mixture used in the ferrihydrite incubation experiments should have stimulated growth of many peat microorganisms involved in different steps of anaerobic organic matter degradation using Fe(III) as terminal electron acceptor. Differences in the gene copy numbers at the end of the incubation suggested that not only did the supplemented substrates support growth but also OM adsorbed or coprecipitated to

the surface of the ferrihydrite minerals. Based on these results, we can infer that the mineral associated OM was at least partially degradable and not totally protected by its association to minerals or by its molecular structure and size. Furthermore, it is likely that a portion of the ferrihydrite associated OM became re-dissolved over the course of the incubation period, which may have facilitated its utilization by the microbes. Thus, the enhanced growth observed in both the *S. oneidensis* and the microbial consortia can be attributed to the presence of additional complex carbon sources in the form of OM-ferrihydrite

complexes compared to pure ferrihydrite.

### 4.1 Mechanisms of OM-ferrihydrite reduction by *S. oneidensis* MR-1 and the microbial consortia

The results of this study demonstrate that ferrihydrite-associated OM stimulates reduction by *S. oneidensis* MR-1 and the microbial consortia, as well as growth. Fe(II) production increased as the C/Fe ratios increased in all incubations. Both the reduction rates and the total amount of Fe(II) produced at the end of incubation (~12 days) increased in the order Fh < adsorbed

OM-ferrihydrite < coprecipitated ferrihydrite. (Fig. 1, Fig. 2, Fig. 7, and Table 1). Samples containing coprecipitated OM-ferrihydrite complexes were more reactive than samples containing adsorbed OM-ferrihydrite complexes with similar OM loadings. These findings support the results presented in previous studies examining the role of humic acids as potential electron shuttles by *S. putrefaciens* under Fe(III) reducing conditions. These studies showed that *S. putrefaciens* was capable of using humic acids as an electron shuttle to enhance Fe(III) reduction as long as the concentration of organic material was

high enough to overcome any slowing in the reduction processes caused by OM sorption to the surface reduction sites of the mineral (Amstaetter et al., 2012; Shimizu et al., 2013). Our study revealed that a water extract of natural soil organic matter, which has lower contents of aromatic groups, and usually lower electron accepting and electron donating capacities than humic acids (Fritzsche et al., n.d.; Piepenbrock et al., 2014) can also provide enough electron shuttling compounds to stimulate ferrihydrite reduction by *S. oneidensis* as well as by the peat microbial consortium. The organic matter used to produce the

OM-ferrihydrites for this study was extracted by $H_2O$ from a Podzol litter layer and showed an electron accepting capacity of EAC = 32 ±1 mmol e$^-$ (mol C)$^{-1}$ and an electron donating capacity of EDC = 24.7± 1.2 mmol e$^-$ (mol C)$^{-1}$ (see WEOM$_{WINTER}$) (Fritzsche et al., n.d.).

Furthermore, since the *S. oneidensis* MR-1 pre-culture used in the present study was grown aerobically and the peat microbial consortia was resuspended in 10 mL anoxic 0.85% NaCl solution following extraction from peat cores, we can

assume no electron shuttling molecules were secreted by the microorganisms themselves prior to inoculating the incubation



experiments. Thus, the absence of any electron shuttles in our pure ferrihydrite incubations might explain the lowest Fe(III) reduction rates out of all the experiments conducted. On the contrary, we suggest the presence of electron shuttling compounds, such as quinones, in the incubations with coprecipitated or adsorbed OM. Quinone moieties have been described as electron shuttling compounds in previous studies (Scott et al., 1998). In our experiments even small contents of OM on ferrihydrite

(44 mg g$^{-1}$, i.e., small molar C/Fe ratios of 0.4) lead to increased reduction rates. This is in contrast to Shimizu et al. (2013), who reported surface passivation at low C/Fe ratios, but increased reduction rates at molar ratios C/Fe > 1.8.A subsequent study by Poggenburg, et al. (2016) reported enhanced Fe(III) reduction rates in incubations containing OM-ferrihydrite complexes with molar C/Fe ratios of 0.9 – 2.1 and using *S. putrefaciens* as model reducer. Here reduction rates differed mostly due to OM-composition, i.e. aromaticity: Lowest reduction rates were found for ferrihydrite coprecipitates produced with

extracellular polymeric substances, while highest reduction rates were found for ferrihydrite coprecipitates produced with a water extract from Podzol litter. The latter is likely comparable in composition to the OM of our experiment. However, absolute reduction rates cannot be compared directly, because Poggenburg et al. (2016) attributed their high rates of Fe(III) reduction partly to the desorption of NOM from coprecipitates due to large concentration of phosphate in their nutrient solutions. In the present study, our C/Fe molar ratios ranged from 0.4 to 4.2, yet we observed an increase in Fe(III) reduction

rates for all incubations containing either coprecipitated or adsorbed OM-ferrihydrite complexes compared to incubations with pure ferrihydrite.

In *Geobacter* sp., direct contact with oxide surfaces is required for Fe(III) reduction. *Geobacter* produces nanowires, typically described as conductive extracellular appendages, which facilitate electron transfer by serving as an electrical conduit to the Fe(III) oxide surface. Conversely, *Shewanella* species are capable of using three primary strategies to allow for electron

transfer between itself and solid Fe(III) oxide surfaces. *Shewanella* has been shown to transfer electrons via direct contact with the oxide surfaces, production of a complexing ligand (i.e. siderophore), or the utilization of endogenously or exogenously produced electron shuttles. A previous study by Eusterhues, et al. (2014) showed the rates of Fe(III) reduction by *G. bremensis* decreased as the concentrations of OM adsorbed or co-precipitated to the surface of ferrihydrite increased (Eusterhues, *et al*. 2014). However, under identical reduction experiments and the same ferrihydrite material, Fe(III) reduction by *G. bremensis*

resulted in considerably higher reduction rates of (0.088-0.148 mM h$^{-1}$) compared to *S. oneidensis* (0.022-0.051 mM h$^{-1}$) and the microbial consortia (0.031-0.058 mM h$^{-1}$) (Table 1, Eusterhues et al., 2014). Although favoured at high concentrations of ferrihydrite-associated OM, electron shuttling is slower than direct electron transfer via nanowires. Similar results were reported in a study by Poggenburg, et al. (2016), in which the rates of Fe(III) reduction by *G. metallireducens* were greatest with pure ferrihydrite compared to ferrihydrite coprecipitates produced with different natural organic materials. These findings

suggest that the reduction rates are not only influenced by organic matter composition and the C/Fe molar ratios, instead they also reflect species dependent capabilities. It was hypothesized that OM may passivate Fe(III) oxide surfaces for direct contact requiring reducers such as *Geobacter*, whereas it may promote reduction by providing exogenous electron shuttles for non-direct contact requiring reducers such as *Shewanella* sp. (Eusterhues et al., 2014). In the current study, we observed a similar behaviour for *S. oneidensis* and the complex microbial consortia extracted from peatland suggesting that non direct-contact



requiring microorganisms dominate reduction in the anoxic, Fe-rich fen, making use of the exogenous electron shuttling compounds of the natural OM (Fig. 7). The similarity of the patterns observed point to the suitability of *S. oneidensis* as model organism to study the effect of OM on ferrihydrite reduction in natural environments.

## 4.2 Potential key players of OM-ferrihydrite reduction

Though the microbial consortia behaves similarly to *S. oneidensis* MR-1 with regards to Fe(III) reduction capacities and growth
over time, the community profiles at the end of incubation did not reveal an enrichment of *Shewanella* species. Differential abundance analyses between adsorbed OM-ferrihydrite incubations and the pure ferrihydrite incubation even revealed reduced abundances for *Shewanella* in case of adsorbed OM-ferrihydrite incubations. The enriched community is indeed comprised of microorganisms capable of reducing Fe(III) oxides, however, *Shewanella* sp. but also and *Geobacter* sp. did not respond to these growth conditions and are noticeably low abundant. But the lack of enrichment of typical dissimilatory Fe(III)-reducer
cannot be explained by the lack of suitable electron donors. Previous work using $^{13}$C-labelled acetate in paddy soils revealed that especially *Geobacter*-related organisms respond to acetate amendment (Hori et al., 2010).

     We saw distinct changes in microbial community structure over time as reflected by alpha- and beta diversity analyses which appeared to be inked to several main drivers shaping the microbial communities in the incubations. The unexpectedly high alpha-diversity observed for the adsorbed OM-ferrihydrite incubation featuring a medium C/Fe ratio (AFhB) is a result
of over-estimating diversity, due to a high frequency of singletons (OTUs that are only represented by single sequences). The overall anaerobic incubation conditions with ferrihydrite offered as terminal electron acceptor and a mixture of the carbon substrates glucose, lactate, and acetate activated only a subset of the original community extracted from peat. Fermentative Enterbacteriaceae (*Ewingella, Rahnella, Serratia*) were overrepresented in the pure ferrihydrite incubation which is reflective of their ability to use glucose as carbon source and available Fe(III) as electron sink. Enterbacteriaceae are known for a mixed
acid fermentation profile similar to the products observed in earlier peat soil incubation studies amended with glucose (Reiche et al., 2008). The question whether Fe(III) can be also used for energy conservation by these bacteria remained unanswered thus far. Early work by Ottow (1970) suggested *Serratia* to be involved in hematite reduction (Ottow, 1970). The strong reduction in their relative abundance in OM-ferrihydrite incubations is presumably a result of Fe(III) being less accessible.

     Community profiles of OM-ferrihydrite incubations differed tremendously from the one of the pure ferrihydrite
incubation. Pseudomonadaceae were highly enriched in all OM-ferrihydrite incubations. The enrichment of Veillonellaceae in adsorbed and coprecipitated OM-ferrihydrite incubations with low C concentrations could be mostly attributed to *Pelosinus*. *Pelosinus* sp. have been characterized as fermentative bacteria which utilize Fe(III) as an electron sink (Shelobolina et al., 2007). These bacteria can use $H_2$ as an electron donor and acetate as a carbon source, and are also capable of fermenting citrate, lactate, and organic impurities. *Pelosinus* has also been shown to reduce small concentrations of anthraquinone-2.6-disulfonate
(AQDS), a synthetic quinone, in the presence of citrate or lactate (Shelobolina et al., 2007). The ability of these microorganisms to use AQDS as an electron sink under reducing conditions might explain their enrichment during our





incubations. Quinone moieties are ubiquitous in soils and peat, thus providing the perfect breeding ground for these microorganisms.

Interestingly, Fe(III)-reduction was previously found to be correlated with the presence of Desulfovibrionaceae, when lactate is available as carbon source (Lentini et al., 2012). This partially matches our observations for incubations with co-precipitated ferrihydrite where *Desulfovibrio* was enriched at low and medium C/Fe ratios. However, *Desulfovibrio* were nearly absent in adsorbed OM-ferrihydrite incubations. Differences in Fe(III) accessibility in adsorbed and coprecipitated OM-ferrihydrite or the higher solubility of coprecipitates might explain this difference in community structure. The presence and ecological role of sulfate-reducing bacteria in the Schlöppnerbrunnen fen has been documented several times (Loy et al., 2004; Pester et al., 2010). This acidic (pH 4), low-sulfate (20-200 µM) peatland shows significant sulfate-reducing capacities (Loy et al., 2004) which is explained by a rapid recycling mechanism (Küsel et al., 2008). Recycling of sulfate can proceed by the aerobic oxidation of sulfide, e.g., in regions where oxygen penetration and anoxic micro-niches overlap (Knorr and Blodau, 2009). The sulfate-reducing activity is mainly attributed to a member of the 'rare biosphere' in the peat, a *Desulfosporosinus* species (Pester et al., 2010). Our study suggests that other members of the sulfate reducers are involved in Fe(III) reduction in this fen to overcome the temporal and/or spatial limitation of the terminal electron acceptor sulfate.

### 4.3 Mineral transformation during reduction (XRD)

X-ray diffraction after reduction showed the presence of remaining ferrihydrite as well as salts which precipitated from the medium. Goethite, other Fe(III) oxides or Fe(II) minerals were not found. The newly formed Fe(II) was either effectively chelated by the present organic material or dissolved Fe concentrations were too low to exceed the solubility product of Fe oxides or other common secondary Fe(II) minerals. The latter seems likely as the reduction rates of the present experiment were in fact low (0.022 – 0.058 mM h$^{-1}$; Table 1). In previous experiments, using the same OM-ferrihydrites, the same medium, but *Geobacter bremensis* as the model Fe(III) reducer, we detected goethite and siderite for samples without OM or only low OM contents (Fh, AFhA, CFhA, CFhB), where reduction rates where high (0.119-0.148 mM h$^{-1}$). Other similar experiments reported no mineral transformation (Henneberry et al., 2012) or observed goethite formation for pure ferrihydrite, while OM-ferrihydrites transformed into green rust and magnetite (Shimizu et al., 2013).

### 4.4 Summary and Conclusions

In the present study mineral-bound soil OM is shown to increase microbial Fe(III) reduction by *S. oneidensis* MR-1 and by a microbial consortia extracted from Schlöppnerbrunnen fen peat cores. The reactivity of ferrihydrites with adsorbed OM differed from ferrihydrites with coprecipitated OM, such that higher extents of Fe(III) reduction were observed for coprecipitated ferrihydrites in comparison to adsorbed ferrihydrites. This can be explained by the smaller crystal size and more defective crystal structure of coprecipitated ferrihydrites compared to normal ferrihydrites (with or without adsorbed organic matter). Additionally, reduction rates were slightly higher for the microbial consortia than for *S. oneidensis* MR-1. The fact that reduction rates for both *S. oneidensis* MR-1 and the microbial consortia increased systematically with increasing



contents of OM in the OM-ferrihydrite complexes tempts us to speculate about how reduction rates are related to electron
transfer mechanisms.  *Shewanella* sp. are capable of reducing Fe(III) oxides using one of three mechanisms known for
reductions, including direct contact, electron shuttles, or production of complexing ligands.  Thus, the enhanced Fe(III)
reduction rates in our incubations led us to the conclusion that under these conditions electron shuttling using exogenous
moieties of the mineral-bound OM is likely the preferred mechanism used for reduction (Fig. 7).  In natural environments,
coprecipitation of OM to mineral surfaces is a common process in nature, thus, we can hypothesize that electron shuttling
mechanisms are the dominant mechanism for microbial Fe(III) reduction in environments with high concentration of OM,
such as soils, sediments, and peat.

Experiments with the microbial consortia extracted from peat showed that the presence of mineral associated OM
influenced the microbial community composition: Whereas *Enterobacteriaceae* and *Burkholderiaceae* dominated in
incubations with pure ferrihydrite, *Pseudomonadaceae* were the dominant microbes present when ferrihydrites were (partly)
covered by OM. When ferrihydrites had low organic matter loadings *Veillonellaceae* occurred in relatively high
concentrations, conversely, *Rhodospirillaceae* are found only in samples with high OM contents. In incubations with OM-
mineral complexes, dominant microbes enriched included *Desulfobacteria* and *Pelosinus* sp., both of which can utilize lactate
and acetate as an electron donor under Fe(III) reducing conditions.

Thus, our observations imply OM coverage does not protect the underlying Fe mineral from microbial reduction in
soils dominated by microorganisms that predominantly use electron shuttles or ligands for Fe(III) reduction. Our study also
revealed that natural soil organic matter on the surface of ferrihydrites provides enough redox active groups to maintain
microbial Fe(III) reduction processes, while the additional carbon in the form of OM enhances overall microbial growth and
shapes the microbial community structure.

*Data availability*. Generated amplicon sequencing data can be found as raw data at NCBI SRA and are available under the
following BioProject accession number: PRJNA391563.

*Author contributions*. K. Eusterhues, K. U. Totsche, and K. Küsel designed the experiments. R.E. Cooper performed the
experiments.  K. Eusterhues synthesized the pure ferrihydrite and OM-ferrihydrite complexes.  C-E Wegner carried out
sequence analyses. R.E. Cooper prepared the manuscript with contributions from all co-authors.

*Acknowledgement*s. Support for this research was kindly provided by the Deutsche Forschungsgemeinschaft (DFG) through
the collaborative research center 1127 Chemical Mediators in Complex Biosystems (ChemBioSys) and the German Centre for
Integrative Biodiversity Research (iDiv) Halle-Jena-Leipzig (FZT 118). We thank Julia Neidhardt, Anke Hädrich, Katy
Pfeiffer, and Gundula Rudolph for help in the laboratory.





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





| | | C | Fe | C/Fe | *S. oneidensis* MR-1 linear fit | | | microbial consortia linear fit | | | *G. bremensis*\* linear fit | | |
|---|---|---|---|---|---|---|---|---|---|---|---|---|---|
| | | | | | k | $r^2$ | Fe(II) at $T_{end}$ | k | $r^2$ | Fe(II) at $T_{end}$ | k | $r^2$ | Fe(II) at $T_{end}$ |
| | | mg g$^{-1}$ | mg g$^{-1}$ | mol mol$^{-1}$ | mM h$^{-1}$ | | % of Fe | mM h$^{-1}$ | | % of Fe | mM h$^{-1}$ | | % of Fe |
| control | Fh | - | 560 | - | 0.022 | 0.98 | 10 | 0.031 | 0.95 | 15 | 0.119 | 0.96 | 59 |
| adsorbed OM | AFhA | 44 | 527 | 0.39 | 0.032 | 0.96 | 15 | 0.037 | 0.97 | 17 | 0.148 | 0.96 | 55 |
| | AFhB | 105 | 468 | 1.04 | 0.038 | 0.98 | 14 | 0.044 | 0.97 | 16 | 0.096 | 0.96 | 35 |
| | AFhD | 181 | 342 | 2.46 | 0.041 | 0.97 | 16 | 0.050 | 0.96 | 19 | 0.088 | 0.94 | 32 |
| coprecip. OM | CFhA | 44 | 498 | 0.41 | 0.038 | 0.97 | 17 | 0.046 | 0.98 | 19 | 0.142 | 0.93 | 66 |
| | CFhB | 98 | 428 | 1.06 | 0.039 | 0.97 | 17 | 0.048 | 0.95 | 20 | 0.142 | 0.97 | 49 |
| | CFhD | 182 | 299 | 2.83 | 0.051 | 0.99 | 20 | 0.058 | 0.97 | 22 | 0.091 | 0.95 | 39 |

\* data from Eusterhues et al., (2014)


**Table 1.** C and Fe concentration and molar C/Fe ratio of ferrihydrite-organic matter associations and results of microbial reduction experiments (reduction rate k and Fe(II) concentration after 296.5 hours of incubation in % of total Fe).







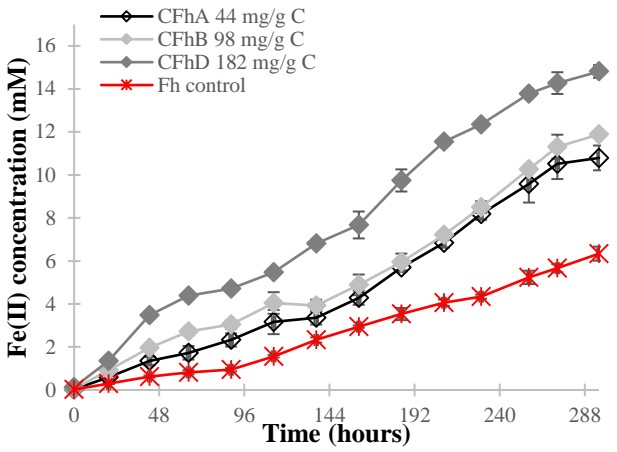

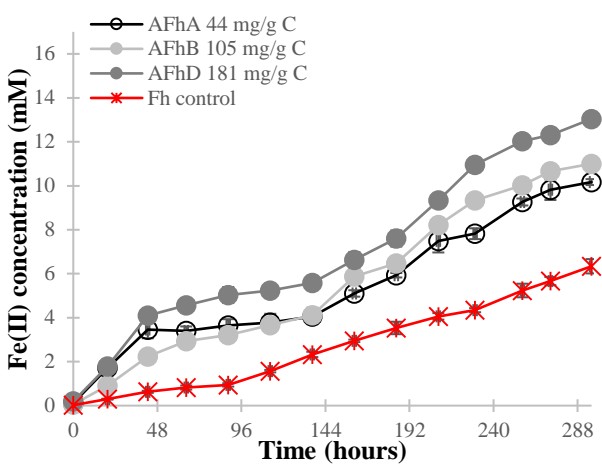

**Figure 1.** Microbial reduction of ferrihydrite (Fh control) and organic matter (OM) ferrihydrite complexes (coprecipitated OM-ferrihydrite: CFh; adsorbed OM-ferrihydrite: AFh) in incubations with *Shewanella oneidensis* MR-1. Error bars represent standard deviation in triplicate incubations.



A) Coprecipitated OM-Ferrihydrite

B) Adsorbed OM-Ferrihydrite

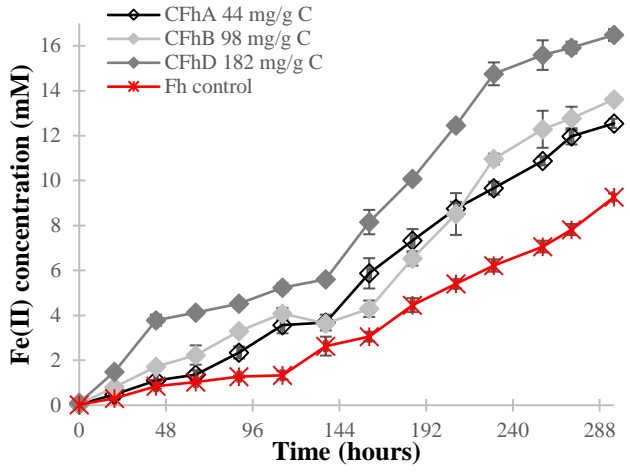

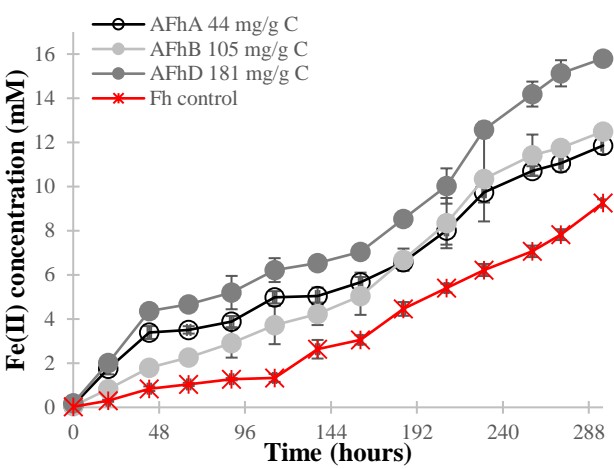

**Figure 2.** Microbial reduction of ferrihydrite and organic matter (OM) ferrihydrite complexes (coprecipitated OM-ferrihydrite: CFh; adsorbed OM-ferrihydrite: AFh) in incubations containing microbial consortia extracted from peatland. Error bars represent standard deviation in triplicate incubations.



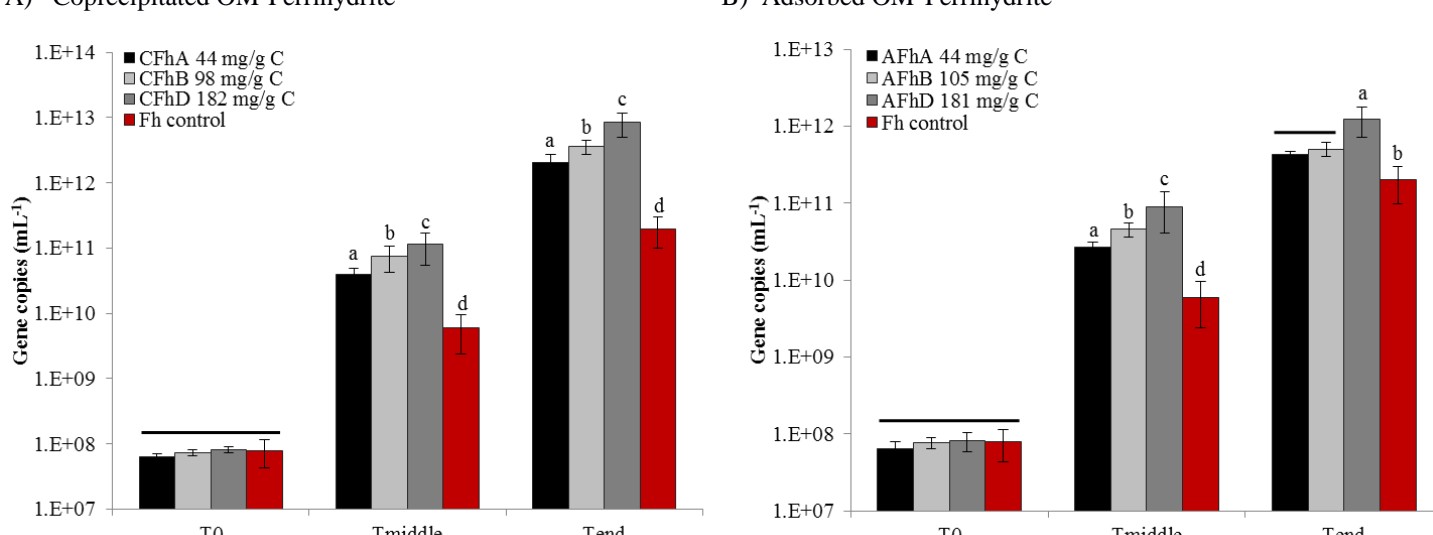

**Figure 3.** Abundances of *S. oneidensis* MR-1 16S rRNA gene in subsamples taken from ferrihydrite incubations containing *S. oneidensis* MR-1 at $T_0$ (0 h), $T_{middle}$ (136 h), $T_{end}$ (296 h). Bars represent mean abundances of triplicate samples from incubations with coprecipitated OM-ferrihydrite (CFh) (A) or adsorbed OM-ferrihydrite (AFh) (B). Each sample was measured in triplicate qPCR reactions with applicable standard deviations. One-way ANOVA was applied to compare individual datasets for statistical significant differences between ferrihydrite incubations. Different letters (a-d) indicate statistically significant differences, while horizontal black lines indicate non-significant differences.




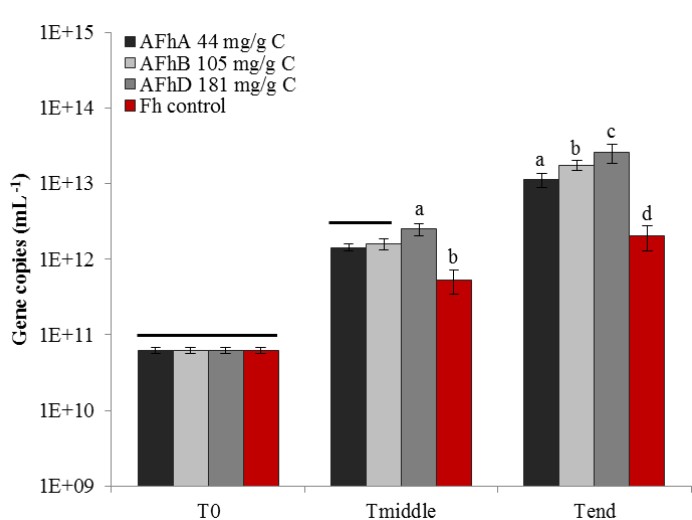

**Figure 4.** Abundances of bacteria l6S rRNA genes in subsamples taken from ferrihydrite incubations containing microbial consortia extracted from peatland at $T_0$ (0 h), $T_{middle}$ (136 h), $T_{end}$ (296 h). Bars represent mean abundances of triplicate samples from incubations with coprecipitated OM-ferrihydrite (CFh) (A) or adsorbed OM-ferrihydrite (AFh) (B). Each sample was measured in triplicate qPCR reactions with applicable standard deviations. One-way ANOVA was applied to compare individual datasets for statistical significant differences between ferrihydrite incubations. Different letters (a-d) indicate statistically significant differences, while horizontal black lines indicate non-significant differences.



**Figure 5.** Microbial community profiling from incubations with coprecipitated OM-ferrihydrite (CFh) or adsorbed OM-ferrihydrite (AFh) compared to microbial communities with pure ferrihydrite (Fh control) at the end of incubation and microbial consortia extracted from peatland at beginning of incubation (peatland microbial extract). Bubble plot indicating the relative abundance of prokaryotic phyla when individual datasets are pooled (A). Bubbles are sized according to respective relative abundances. The ten most abundant phyla are highlighted by color. Grey bubbles refer to less abundant phyla. Family-





level taxonomic profiles are given for individual datasets. Family-level groups showing abundances less than 1%, as well as unclassified OTUs are summarized as "Unclassified/Other"(B). Overrepresented genus-level groups were identified by negative binomial distributions. Tests were carried using ferrihydrite complexation as decisive parameter. Multiple inference correction was done according to Benjamini-Hochberg (1995) (Benjamini and Hochberg, 1995). *p*-values less than 0.05 were considered significant (C).The color scheme in panels B and C is according to panel A. Family-level groups belonging to the

Firmicutes are for instance shown in shades of orange in case of panel B. In panel B, AFhA-C refers to increasing TOC concentrations (44 mg/g C, 105 mg/g C, 181 mg/g C) in adsorbed OM-ferrihydrite incubations and CFhA-C refers to increasing TOC concentrations (TOC: 44 mg/g C, 98 mg/g C, 182 mg/g C) in incubations with coprecipitated ferrihydrite.








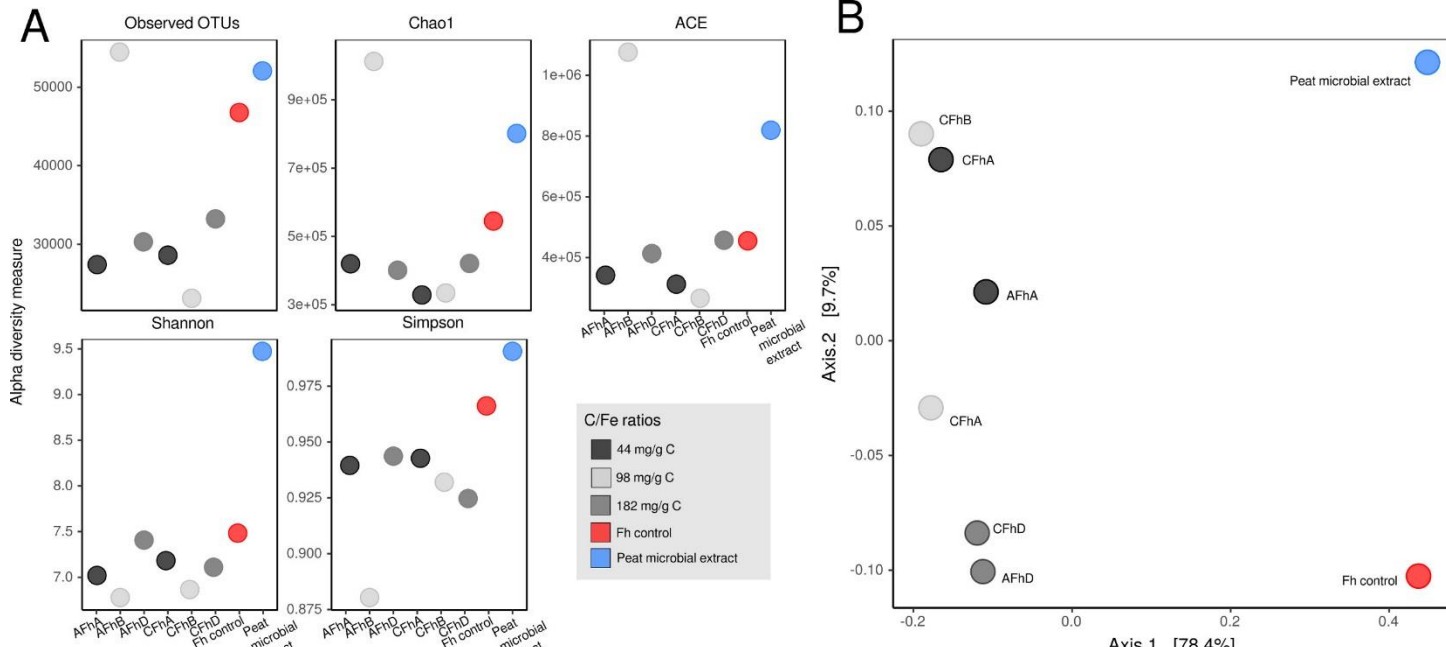

**Figure 6.** Alpha- and beta-diversity analyses. Alpha diversity analysis was done using various indices indicative for diversity (Observed OTUs, Chao1, ACE) and evenness (Shannon, Simpson) (A). Alpha diversity in general describes within dataset

diversity. Observed OTUs refers to OTU counts based on the OTU definition given in the materials and methods. Chao1 and ACE (Chao, 1984; Lee and Chao, 1994), are two indices that put special emphasis on rare taxa to extrapolate the real number of OTUs based on observed OTUs. The Shannon index (Shannon, 1948) describes the entropy in a community profile, and reaches its maximum when taxonomic groups are evenly distributed in a dataset. The Simpson index (Simpson, 1949) refers to the probability that two randomly sampled sequences from a dataset belong to the same OTU. Beta diversity (= differences

in diversity between samples) analysis was based on Jensen-Shannon divergences (Koren et al., 2013) (B).




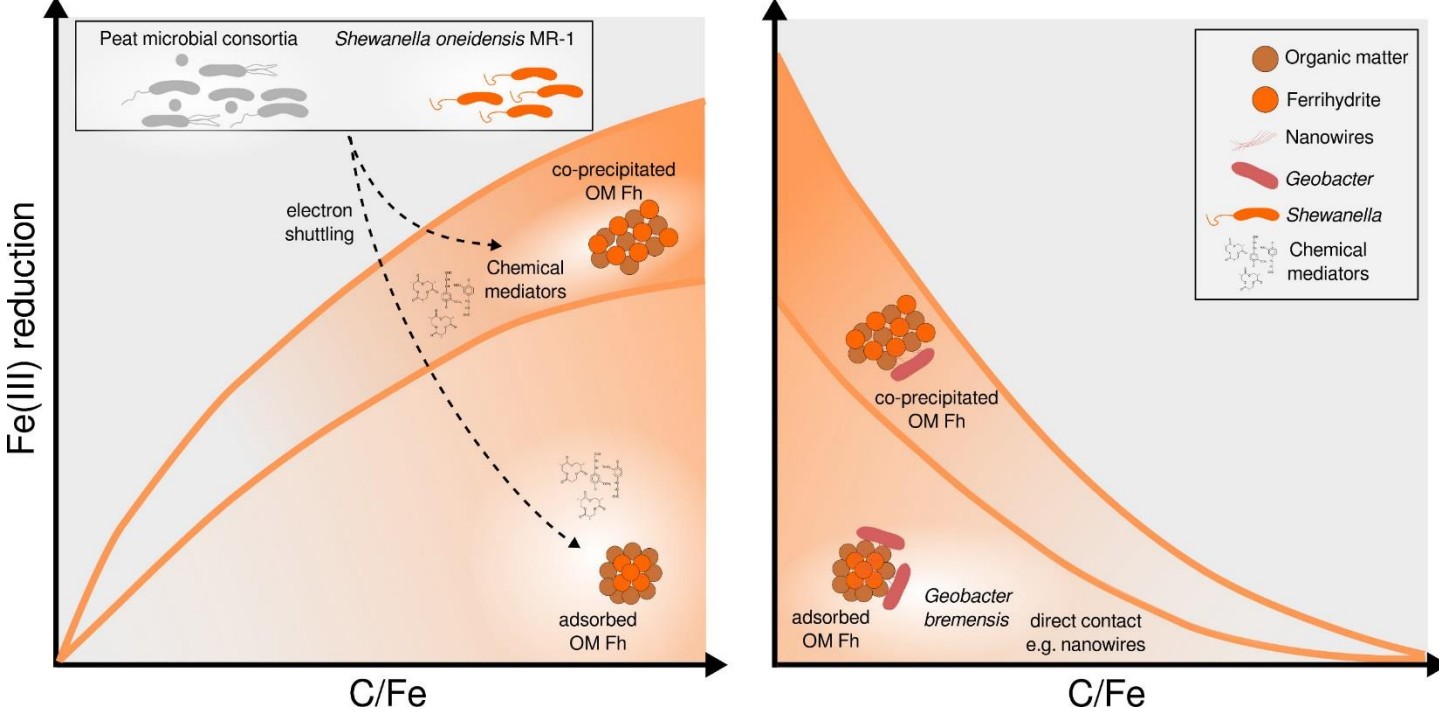

**Figure 7.** Microbial reduction of ferrihydrite with both adsorbed and coprecipitated OM by Shewanella oneidensis MR-1 and the microbial community consortia extracted from Schlöppnerbrunnen peat samples. For comparison, microbial reduction of ferrihydrite with both adsorbed and coprecipitated OM by *Geobacter bremensis* as described in Eusterhues, et al. 2014 is depicted in the right panel.