# Peer review of "Ferrihydrite associated organic matter (OM) stimulates reduction by *Shewanella oneidensis* MR-1 and a complex microbial consortia"

_Biogeosciences, 2017_

## Referee Comment (RC1) · S. Hayes (Referee) · 2 Aug 2017

Review of: Cooper et al., 2017

Manuscript ID: bg-2017-270

Reviewer: Sarah Hayes

Recommendation: Minor Revisions

Summary: Cooper et al. is a thoughtfully designed and compares the rate of microbial iron reduction by Shewanella oneidensis MR-1 and a natural microbial isolate from an Fe-rich fen as a function of organic matter concentration and association with ferrihudrite (sorbed vs coprecipitated). The nature of mineral-organic matter interactions in the natural environment is an active area of research and the influence of nanoscale mineral and organic matter associations on microbial behavior is very timely, interesting, and well-suited for publications in Biogeosciences. The study is very clearly outlined and easy to understand with surprisingly similar rates of Fe reduction by the Shewanella oneidensis MR-1 and a fen microbial isolate. The manuscript is clearly written with a few typos that do not distract from the meaning.

General comments:

Overall, this is a lovely manuscript. The discussion section could use a little bit of streamlining to really highlight the important contributions a bit better.

- It seems like the bulk of the section 4.0 could be redistributed into the other sections of the paper

- Section 4.3 could probably be absorbed by the results section. And any mention in the discussion might be best integrated into another section.

- The conclusions section could also be streamlined to quickly revisit the key findings of the study.

Specific comments:

L11: should be "this study investigated to what extent"

L14: should be "OM content" not "contents"

L48: should be "coprecipitation is the more common process"

L49-51: This sentence isn't super clear. Consider breaking it into two sentences or maybe just clarifying the end, something like "reducing agents potentially differ between ferrihydrite precipitated in the presence of OM and purely adsorbed OM.

L61: Again, this sentence isn't quite right. Consider something like: "dissolved humic

acid and the mineral/humic acid ratios have been shown by some studies to increase Fe (III) reduction rates (ref) while other studies have not reproduced this result (ref)." But I'm not sure it keeps the meaning of what you are trying to communicate.

L85: Consider revising to: "Further, this study found that reduction rates by Shewanella were correlated. . ."

L86-97: This section could flow a bit better. All the pieces are there, but it feels disjointed.

L94-97: This is an excellent way to close this paragraph!

L105-7: I think this surprising result (which is way cool!) should be moved to later in the paper, maybe the results section. It doesn't really fit here in the introduction.

L136: Add the information about the organic matter here (from lines L146-147 and from the discussion L379-82). Then maybe make the rest of the paragraph here with the information about the soil organic matter extraction first and then

L152: Should be "These cultures were. . ."

L326: Add mineral formulas to the list of minerals observed.

L331-4: This sentence is a little difficult to understand, please reword to improve clarity.

L352: I'm not sure of your exact meaning, please consider revising to something similar to these suggestions, depending on your meaning: "amendment with each of these substrates stimulates Fe(III) reduction. . ." or "amendment with all of these substrates stimulates Fe(III) reduction. . ."

L356: Consider "the substrate mixture used in these ferrihydrite. . ."

L329-345: Would some or all of this material be better suited for introductory material? If not, try to make a stronger discussion point.

L358-365: It seems like there is a topic change here that feels disjointed. The summary

comments beginning on L 360 seem like they should precede the content earlier in the paragraph and are important to discuss in their own right.

L367: Consider: "OM stimulates growth and Fe(III) reduction by S. oneidensis MR-1 and the microbial consortia."

L373-4: Consider revising: "reducing conditions, which showed that S. putrefaciens was capable of using humic acids as an electron shuttle to enhance Fe(III) reduction. However, enhanced Fe reduction was only observed when the concentration of organic..."

L377: Consider revising: "which has lower aromatic moieties, and...

L378: Consider revising: "electron shuttling capacity to..."

L379-82: I think the info about the organic matter seems like it should be in the methods section.

L383-386: seems like these details should also be in the methods, although they are also important here.

L394: The colon should probably be a period.

L397: The comma between "directly" and "because" does not seem necessary.

L405: Consider revising: "oxide surfaces: via direct contact..."

L406: The "or" should be an "and"

L428: Consider revising "Fe(III) oxides. However, ..."

L429: "abundant" should be "abundance"

L433: "inked" should be "linked"

L441: Should be "the question of whether"

L441: consider revising: "bacteria remains unanswered."

L442: the citation at the end of the sentence is redundant of in text citation.

L474: Should be "formation from pure..."

L480-482: This is kinda new information, which should be mentioned earlier (maybe with the other content in section 4.3). Further, you need a citation because this study does not provide any direct evidence to support this claim (which is likely the case).

---

## Referee Comment (RC2) · Anonymous Referee #2 · 9 Aug 2017

General Comments:

The manuscript is well written and the main points come across easily. In the introduction, the authors recognize a number of studies that analyze the influence of organic matter (OM) on Fe(III) reduction kinetics and Fe-mineral transformations. These studies have been carried out under abiotic conditions or with pure cultures of well-known Fe-reducers such as Geobacter and Shewanella. This study focuses on (1) defining how a natural microbial consortium influences the reduction of Fe(III)-OM complexes and (2) determining microbial community changes under anaerobic Fe-reducing conditions. These findings will significantly impact our understanding of Fe-OM interactions

under environmentally relevant conditions.

Specific Comments:

Introduction Line 33: should be "The majority of which is dispersed.." Line 44: should be "The coprecipitation of OM with Fe results in..." Line 49-51: This sentence is confusing. It seems like you are trying to make two separate arguments (1) coprecipitated ferrihydrite has different properties than pure ferrihydrite, and (2) coprecipitated ferrihydrite has different reactivity compared to ferrihydrite with sorbed OC. Maybe split into two sentences? Line 74-93: This paragraph has important information that you reply on in your discussion. Lines 94-97 clearly state the overall findings, but I found it a bit difficult to follow lines 74-93. Line 101: should be "is the dominant electron accepting process"

Methods Line 110-112: This first sentence is confusing. Line 117: what is Corg? Line 129: This first sentence seems like it should be last sentence of the previous paragraph. Line 130-133: These sentences are out of place. I don't think you need to talk about DNA extraction or PCR here. Maybe move them to the "DNA extractions" and "Quantitative PCR" sections. Line 146: Move this up so that the reader knows that you performed the forest floor extract solutions. Line 149-156: You may want to add this paragraph to a section called "Preparation of pure cultures". Line 156: I'm not sure why you write about "Defined freshwater medium" here if you also have a much more detailed description in lines 162-167. Line 176: If you are not showing any data (not even in the supplemental information), then why mention that you did this? 179: Did you also collect XRD spectra of the starting materials? Line 199: move this sentence to the end so that it follows the order in which things were done. It will also make it easier for the reader to understand that you did the PCR and prepared the libraries yourself and just did the sequencing at the LGC Genomic GmbH. Line 224: Did you normalize the data using qiime or any other method? If you did, you should mention this because it is important when comparing the abundance of different taxa across various samples.

Results The order of these sections has a great flow. However, I would consider merging the Results and Discussion sections into one section. This will avoid many repetitive statements. Section 3.1: The reduction rates stated in this paragraph are not the same as the ones in Table1. Section 3.3: When reading this section and flipping to the figures, I found it difficult to determine if I was looking at the correct figure. Please double check the Figure numbers that you refer to in the text. For example, in line 285 you mention beta diversity and Figure 5b and I think you meant to reference Figure 6b.

Discussion Line 373: Add reference. Line 394: ":" should be "." Line 404: Add reference. Line 405: Add reference. Line 435: Have you considered removing singletons and see what the data looks like? Section 4.3: Include mineral formulas and consider citing the following paper: Influence of Coprecipitated Organic Matter on Fe2+(aq)-Catalyzed Transformation of Ferrihydrite: Implications for Carbon Dynamics Chunmei Chen, Ravi Kukkadapu, and Donald L. Sparks Environmental Science & Technology 2015 49 (18), 10927-10936 DOI: 10.1021/acs.est.5b02448 Line 480: Where is the evidence of this finding? Maybe consider placing this information in the section about mineral composition. Line 500-503: This last sentence is not clear.

References Consider citing more recent studies such as: Influence of Coprecipitated Organic Matter on Fe2+(aq)-Catalyzed Transformation of Ferrihydrite: Implications for Carbon Dynamics Chunmei Chen, Ravi Kukkadapu, and Donald L. Sparks Environmental Science & TechnologyÂă2015Âă49Âă(18), 10927-10936 DOI: 10.1021/acs.est.5b02448

Dinesh Adhikari, Qian Zhao, Kamol Das, Jacqueline Mejia, Rixiang Huang, Xilong Wang, Simon R. Poulson, Yuanzhi Tang, Eric E. Roden, Yu Yang, Dynamics of ferrihydrite-bound organic carbon during microbial Fe reduction, Geochimica et Cosmochimica Acta, Volume 212, 2017, Pages 221-233, ISSN 0016-7037, http://dx.doi.org/10.1016/j.gca.2017.06.017. (http://www.sciencedirect.com/science/article/pii/S0016703717303666)

Weinan Pan, Jinjun Kan, Shreeram Inamdar, Chunmei Chen, Donald Sparks, Dissimilatory microbial iron reduction release DOC (dissolved organic carbon) from carbon-ferrihydrite association, Soil Biology and Biochemistry, Volume 103, 2016, Pages 232-240, ISSN 0038-0717, http://dx.doi.org/10.1016/j.soilbio.2016.08.026. (http://www.sciencedirect.com/science/article/pii/S0038071716302115) Keywords: Carbon sequestration; DOC-Mineral stability; Shewanella oneidensis MR-1; Dissimilatory iron reduction; Anaerobic conditions

Figures Consider merging Figures 1 and 2 into a 4-panel Figure. This will allow the reader to compare the results obtained from pure cultures and natural inoculum.

Consider merging Figures 3 and 4 into a 4-panel Figure (make sure the y-axis is the same for all). This will allow the reader to compare the results obtained from pure cultures and natural inoculum. You may also want to consider changing T0, T middle and Tend to day 0, day X and day 288. This will keep the axis consistent with the other figures.

Figure 5C. The results illustrated in this Figure were not clear to me. Perhaps state the taxonomic level of the x-axis? Or add different shapes to differentiate between CFh vs Fh and AFh vs Fh?

Figure 6. I understand the importance of using various alpha diversity parameters. However, I am not sure illustrating them all in a main figure is necessary. Perhaps choose one parameters and talk about that one in the paper. The rest could go in the supplemental materials.

---

## Author Comment (AC1) · 6 Sep 2017

**Author's responses to referee #1 (Sarah Hayes) comments on Cooper, et al (2017), 'Ferrihydrite associated organic matter (OM) stimulates reduction by *Shewanella oneidensis* MR-1 and a complex microbial consortia,' *Biogeosciences Discuss.*, doi.org/10.5194/bg-2017-270**

**Reviewer: Sarah Hayes**

Summary: Cooper et al. is a thoughtfully designed and compares the rate of microbial iron reduction by Shewanella oneidensis MR-1 and a natural microbial isolate from an Fe-rich fen as a function of organic matter concentration and association with ferrihydrite (sorbed vs coprecipitated). The nature of mineral-organic matter interactions in the natural environment is an active area of research and the influence of nanoscale mineral and organic matter associations on microbial behavior is very timely, interesting, and well-suited for publications in Biogeosciences. The study is very clearly outlined and easy to understand with surprisingly similar rates of Fe reduction by the Shewanella oneidensis MR-1 and a fen microbial isolate. The manuscript is clearly written with a few typos that do not distract from the meaning.

**We would like to thank referee #1, Sarah Hayes, for taking the time to review our manuscript and for her feedback and supportive comments. Your feedback will undoubtedly improve the quality of the manuscript. Below we provide answers to all comments.**

General comments:

Overall, this is a lovely manuscript. The discussion section could use a little bit of streamlining to really highlight the important contributions a bit better.

**Thanks for the positive feedback!**

-It seems like the bulk of the section 4.0 could be redistributed into the other sections of the paper.

**This section was reorganized/rewritten in a way that the information provided can now stand alone as an independent section instead of needing to be re-distributed to other sections of the manuscript. Please note, this is also addressed in our response to your comments regarding Line 329-345, 358-365 on the next page.**

-Section 4.3 could probably be absorbed by the results section. And any mention in the discussion might be best integrated into another section.

**Section 4.3 was removed from the discussion section. This information was integrated into the results section 3.4 as suggested.**

-The conclusions section could also be streamlined to quickly revisit the key findings of the study.

**We did some changes to the Summary and Conclusions (now section 4.3) to better streamline, however, we feel the way we presented our summary and overall conclusions in this section is effective in getting our message across and would prefer not to make significant changes to the text here.**

Specific comments:

L11: should be "this study investigated to what extent"

**Corrected (Line 11).**

L14: should be "OM content" not "contents"

**Corrected (line 14).**

L48: should be "coprecipitation is the more common process"

**Corrected (line 48).**

L49-51: This sentence isn't super clear. Consider breaking it into two sentences or maybe just clarifying the end, something like "reducing agents potentially differ between ferrihydrite precipitated in the presence of OM and purely adsorbed OM.

**This sentence was modified for clarity. The following sentence was added: 'The different properties of pure ferrihydrite and OM-ferrihydrite coprecipitates may lead to different behaviors during microbial reduction. Due to their smaller crystal size and more defective crystal structure, coprecipitates might faster dissolve. The associated organic material will change the mineral´s surface properties, e.g., the**

**surface charge, with consequences for the accessibility of Fe(III) to microbes, redox-active shuttling compounds, or extracellular enzymes.' (Line 49-52)**

L61: Again, this sentence isn't quite right. Consider something like: "dissolved humic acid and the mineral/humic acid ratios have been shown by some studies to increase Fe (III) reduction rates (ref) while other studies have not reproduced this result (ref)." But I'm not sure it keeps the meaning of what you are trying to communicate.

**This line was modified as suggested (Line 65-66).**

L85: Consider revising to: "Further, this study found that reduction rates by Shewanella were correlated..."

**This sentence was revised as suggested (Line 90).**

L86-97: This section could flow a bit better. All the pieces are there, but it feels disjointed.

**Modifications to this section were included to improve the flow (Line 94-101)**

L94-97: This is an excellent way to close this paragraph!

**Thank you!**

L105-7: I think this surprising result (which is way cool!) should be moved to later in the paper, maybe the results section. It doesn't really fit here in the introduction.

**We considered your suggestion to remove this sentence, but we prefer to leave this as the conclusion to the introduction.**

L136: Add the information about the organic matter here (from lines L146-147 and from the discussion L379-82). Then maybe make the rest of the paragraph here with the information about the soil organic matter extraction first and then

**Information about organic matter moved to Line 144.**

L152: Should be "These cultures were..."

**Corrected (Line 167).**

L326: Add mineral formulas to the list of minerals observed.

**Mineral formulas added to the list of minerals observed in XRD spectra (Line 360-361).**

L331-4: This sentence is a little difficult to understand, please reword to improve clarity.

**This section was modified to improve clarity. Please see section 3.4.**

L352: I'm not sure of your exact meaning, please consider revising to something similar to these suggestions, depending on your meaning: "amendment with each of these substrates stimulates Fe(III) reduction..." or "amendment with all of these substrates stimulates Fe(III) reduction..."

**Sentence modified as suggested ('amendment with each of these substrates stimulates Fe(III) reduction…') (Line 401-402).**

L356: Consider "the substrate mixture used in these ferrihydrite..."

**Sentence modified to reflect your suggestions (line 405-406).**

L329-345: Would some or all of this material be better suited for introductory material? If not, try to make a stronger discussion point. L358-365: It seems like there is a topic change here that feels disjointed. The summary comments beginning on L 360 seem like they should precede the content earlier in the paragraph and are important to discuss in their own right.

**Thanks for pointing out that this first section of the Discussion was a bit disjointed. We took some time to read through and decided the best way to address this was to rewrite/reorganize the information presented in an effort to better link the first paragraph with the overall outcome of our study. The 'second paragraph' is now divided into two separate paragraphs which, in our opinion, are easier and more clear to read and provide a better link to the first paragraph as well as the subsequent sections in the discussion (Line 397-420).**

L367: Consider: "OM stimulates growth and Fe(III) reduction by S. oneidensis MR-1 and the microbial consortia."

**Sentence modified as suggested (line 442-443).**

L373-4: Consider revising: "reducing conditions, which showed that S. putrefaciens was capable of using humic acids as an electron shuttle to enhance Fe(III) reduction. However, enhanced Fe reduction was only observed when the concentration of organic..."

**Sentence modified as suggested. See lines 451-453.**

L377: Consider revising: "which has lower aromatic moieties, and...

**Sentence modified as suggested (Line 456).**

L378: Consider revising: "electron shuttling capacity to..."

**Sentence modified as suggested (Line 456-457).**

L379-82: I think the info about the organic matter seems like it should be in the methods section.

**Information removed from Section 4.1.**

L383-386: seems like these details should also be in the methods, although they are also important here.

**Sentence modified so that sufficient information remains here (Line 463), and more detailed information can be found in the methods (Section 2.2).**

L394: The colon should probably be a period.

**Corrected (Line 473).**

L397: The comma between "directly" and "because" does not seem necessary.

**Corrected (Line 476).**

L405: Consider revising: "oxide surfaces: via direct contact..."

**Sentence modified as suggested (line 484).**

L406: The "or" should be an "and"

**Corrected (line 485).**

L428: Consider revising "Fe(III) oxides. However, ..."

**Sentence modified as suggested (Line 508).**

L429: "abundant" should be "abundance"

**Corrected (Line 509).**

L433: "inked" should be "linked"

**Corrected (line 513).**

L441: Should be "the question of whether"

**Corrected (Line 521).**

L441: consider revising: "bacteria remains unanswered."

**Sentence modified as suggested (Line 4521-522).**

L442: the citation at the end of the sentence is redundant of in text citation.

**Citation at the end of the sentence removed (Line 522).**

L474: Should be "formation from pure..."

**Corrected (Line 372). Note, information in section 4.3 incorporated into results section 3.4.**

L480-482: This is kinda new information, which should be mentioned earlier (maybe with the other content in section 4.3). Further, you need a citation because this study does not provide any direct evidence to support this claim (which is likely the case).

**This information is now introduced in Section 4.1 (Line 447-449). References were added to support this claim.**

---

## Author Comment (AC2) · 6 Sep 2017

**Author's responses to comments from referee #2 on Cooper, et al (2017), 'Ferrihydrite associated organic matter (OM) stimulates reduction by *Shewanella oneidensis* MR-1 and a complex microbial consortia,' *Biogeosciences Discuss.*, doi.org/10.5194/bg-2017-270**

**Anonymous Referee #2**

General Comments:

The manuscript is well written and the main points come across easily. In the introduction, the authors recognize a number of studies that analyze the influence of organic matter (OM) on Fe(III) reduction kinetics and Fe-mineral transformations. These studies have been carried out under abiotic conditions or with pure cultures of well-known Fe-reducers such as Geobacter and Shewanella. This study focuses on (1) defining how a natural microbial consortium influences the reduction of Fe(III)-OM complexes and (2) determining microbial community changes under anaerobic Fe-reducing conditions. These findings will significantly impact our understanding of Fe-OM interactions under environmentally relevant conditions.

**We thank our second referee, anonymous referee #2, for taking the time to review our manuscript and for his/her feedback and helpful comments. Your comments will definitely enhance the quality of the manuscript. Below you will find our responses to all comments.**

Specific Comments:

Introduction

Line 33: should be "The majority of which is dispersed.."

**Corrected (Line 33).**

Line 44: should be "The coprecipitation of OM with Fe results in..."

**Corrected (Line 44).**

Line 49-51: This sentence is confusing. It seems like you are trying to make two separate arguments (1) coprecipitated ferrihydrite has different properties than pure ferrihydrite, and (2) coprecipitated ferrihydrite has different reactivity compared to ferrihydrite with sorbed OC. Maybe split into two sentences?

**Sentence revised for clarification as follows: 'The different properties of pure ferrihydrite and OM-ferrihydrite coprecipitates may lead to different behaviors during microbial reduction. Due to their smaller crystal size and more defective crystal structure, coprecipitates might faster dissolve. The associated organic material will change the mineral´s surface properties, e.g., the surface charge, with consequences for the accessibility of Fe(III) to microbes, redox-active shuttling compounds, or extracellular enzymes.' (Line 49-52).**

Line 74-93: This paragraph has important information that you reply on in your discussion. Lines 94-97 clearly state the overall findings, but I found it a bit difficult to follow lines 74-93.

**Modifications to this section were included to improve the flow (Line 94-101)**

Line 101: should be "is the dominant electron accepting process"

**Corrected (Line 104).**

Methods

Line 110-112: This first sentence is confusing.

**Sentence modified for clarification as follows: 'Peat cores were obtained from the Schlöppnerbrunnen fen (Northern Bavaria, Germany; 50°07'55''N, 11°52'52''E) using a Pürkheimer soil corer in May 2016. This minerotrophic, slightly acidic (pH ~5) fen has been previously described in detail (Blodau et al., 2004; Eusterhues et al., 2014; Hausmann et al., 2016; Küsel et al., 2008; Loy et al., 2004; Pester et al., 2012).' (Line 119-121).**

Line 117: what is Corg?

**'Solid organic carbon' added for clarification (Line 127).**

Line 129: This first sentence seems like it should be last sentence of the previous paragraph.

**The sentence is now the concluding sentence of the previous paragraph as suggested (Line 139-140).**

Line 130-133: These sentences are out of place. I don't think you need to talk about DNA extraction or PCR here. Maybe move them to the "DNA extractions" and "Quantitative PCR" sections.

**The descriptions regarding DNA extraction and subsequent qPCR analyses were moved to section 2.5 (Line 196-202).**

Line 146: Move this up so that the reader knows that you performed the forest floor extract solutions.

**Information about forest floor extract solution moved to Line 144 (detailed descriptions can be found in Esterhues, et al. 2014).**

Line 149-156: You may want to add this paragraph to a section called "Preparation of pure cultures".

**We appreciate the suggestion, however, we have decided to keep the information regarding pure culture preparation and the subsequent incubation set-up with both *S. oneidensis* and the microbial consortia extracted from peat in one section instead of separating the information into separate sections (2.4 Microbial OM ferrihydrite reduction experiments).**

Line 156: I'm not sure why you write about "Defined freshwater medium" here if you also have a much more detailed description in lines 162-167.

**Sentence modified to prevent repetition of detailed description as follows: 'Cells were washed twice with a defined medium, resuspended in 2 ml of medium and diluted to a final concentration of $2 \times 10^5$ cells $ml^{-1}$ in culture tubes, containing a defined medium.' (Line 168-170).**

Line 176: If you are not showing any data (not even in the supplemental information), then why mention that you did this?

**Secondary mineral formation during reduction is of interest in soil and sediment research. Many previous studies address this question. We therefore believe it is important to mention that we also paid attention to this aspect, although we did not find any new minerals. However, a table or a figure with 21 very similar diffractogramms does not seem necessary to us to document our findings.**

179: Did you also collect XRD spectra of the starting materials?

**Yes, XRD spectra was collected from the starting material. Section 2.10 (Line 264-265) and 3.4 (Line 360-361) were both modified to reflect this.**

Line 199: move this sentence to the end so that it follows the order in which things were done. It will also make it easier for the reader to understand that you did the PCR and prepared the libraries yourself and just did the sequencing at the LGC Genomic GmbH.

**PCR, library preparation, and sequencing was all done by LGC Genomic GmbH. Including this type of detailed information (i.e. primers, etc.) in the materials and methods is commonly done. Sentence added to this section to make it clear that LCG did the PCR, library prep, and subsequent sequencing.**

Line 224: Did you normalize the data using qiime or any other method? If you did, you should mention this because it is important when comparing the abundance of different taxa across various samples.

**Yes, the data was normalized using qiime. An additional sentence was added to explain how differentially abundant taxonomic groups were identified after the metadata was assembled for downstream analysis in qiime (Line 252-256).**

Results The order of these sections has a great flow. However, I would consider merging the Results and Discussion sections into one section. This will avoid many repetitive statements.

**We opted to keep the results and discussion sections as independent section and based on comments you provided as well as the other reviewer we hope the repetitiveness is minimized while maintaining the flow of the results and subsequent discussion section.**

Section 3.1: The reduction rates stated in this paragraph are not the same as the ones in Table1.

**Reduction rates stated in paragraph corrected to reflect those found in Table 1.**

Section 3.3: When reading this section and flipping to the figures, I found it difficult to determine if I was looking at the correct figure.

Please double check the Figure numbers that you refer to in the text. For example, in line 285 you mention beta diversity and Figure 5b and I think you meant to reference Figure 6b.

**Figure numbers double-checked. Thanks for pointing this out. Additionally, it's important to note that all figures were re-numbered due to merging of Figures 1 and 2, 3 and 4.**

Discussion
Line 373: Add reference.

**References added (Line 452).**

Line 394: ":" should be ".".

**Corrected (Line 473).**

Line 404: Add reference. Line 405: Add reference.

**References added (Line 486).**

Line 435: Have you considered removing singletons and see what the data looks like?

**Yes, we did consider this and added now the following sentences to section 2.8 (Microbiome profiling): 'After alpha diversity analyses, the dataset was stripped by removing OTUs that were represented by less than three sequences. Relative abundances were calculated and standardized based on median sequencing depth across all samples. Datasets were not rarefied to prevent a loss of sensitivity in downstream sample-wise comparisons (McMurdie and Holmes, 2014). The dataset was stripped by removing OTUs that were only identified in single samples (i.e. singletons).' (Line 243-247). Additionally, the impact on the inclusion of singletons is addressed in Lines 513-515.**

Section 4.3: Include mineral formulas and consider citing the following paper: Influence of Coprecipitated Organic Matter on Fe2+(aq)Catalyzed Transformation of Ferrihydrite: Implications for Carbon Dynamics Chunmei Chen, Ravi Kukkadapu, and Donald L. Sparks Environmental Science & Technology 2015 49 (18), 10927-10936 DOI: 10.1021/acs.est.5b02448

**Mineral formulas added to the list of minerals observed in XRD spectra (Line 360-363). Citation also added as suggested.**

Line 480: Where is the evidence of this finding? Maybe consider placing this information in the section about mineral composition.

**We introduced this information Section 4.1 (Line 447-453) and references were added to support this claim.**

Line 500-503: This last sentence is not clear.

**The last sentence was rewritten for clarity as follows: 'This study also revealed that natural soil OM on the surface of ferrihydrites provides enough redox active groups to maintain microbial Fe(III) reduction processes, instead of passivating the surface. While at the same time the OM enhanced overall microbial growth and shaped the microbial community structure.' (Line 580-585).**

References
Consider citing more recent studies such as: Influence of Coprecipitated Organic Matter on Fe2+(aq)-Catalyzed Transformation of Ferrihydrite: Implications for Carbon Dynamics Chunmei Chen, Ravi Kukkadapu, and Donald L. Sparks Environmental Science & Technology 2015 49 (18), 10927-10936 DOI: 10.1021/acs.est.5b02448

Dinesh Adhikari, Qian Zhao, Kamol Das, Jacqueline Mejia, Rixiang Huang, Xilong Wang, Simon R. Poulson, Yuanzhi Tang, Eric E. Roden, Yu Yang, Dynamics of ferrihydrite-bound organic carbon during microbial Fe reduction, Geochimica et Cosmochimica Acta, Volume 212, 2017, Pages 221-233, ISSN 0016-7037, http://dx.doi.org/10.1016/j.gca.2017.06.017.(http://www.sciencedirect.com/science/article/pii/S0016703717303666)

Weinan Pan, Jinjun Kan, Shreeram Inamdar, Chunmei Chen, Donald Sparks, Dissimilatory microbial iron reduction release DOC (dissolved organic carbon) from carbon-ferrihydrite association, Soil Biology and Biochemistry, Volume 103, 2016, Pages 232-240, ISSN 0038-0717,

http://dx.doi.org/10.1016/j.soilbio.2016.08.026.
(http://www.sciencedirect.com/science/article/pii/S0038071716302115) Keywords: Carbon sequestration; DOC-Mineral stability; Shewanella oneidensis MR-1; Dissimilatory iron reduction; Anaerobic conditions

**The first two suggested references (Chen, et al. and Adhikari, et al.) were added to the manuscript. Thank you for pointing out that they were missing.**

Figures Consider merging Figures 1 and 2 into a 4-panel Figure. This will allow the reader to compare the results obtained from pure cultures and natural inoculum.

**Figures 1 and 2 combined into a single 4-panel figure (see Figure 1). Additionally, we modified the figure caption to reflect the information presented in the merged figure.**

Consider merging Figures 3 and 4 into a 4-panel Figure (make sure the y-axis is the same for all). This will allow the reader to compare the results obtained from pure cultures and natural inoculum. You may also want to consider changing T0, T middle and Tend to day 0, day X and day 288. This will keep the axis consistent with the other figures.

**Figures 3 and 4 combined into a single 4-panel figure (see Figure 2). Figure caption modified to reflect information presented in the merged figure.**

Figure 5C. The results illustrated in this Figure were not clear to me. Perhaps state the taxonomic level of the x-axis? Or add different shapes to differentiate between CFh vs Fh and AFh vs Fh?

**The taxonomic level is for each portion of Figure 3 (previously Figure 5) is described in the figure caption (i.e. 3A – prokaryotic phyla level; 3B – family level; 3C – genus level).**

Figure 6. I understand the importance of using various alpha diversity parameters. However, I am not sure illustrating them all in a main figure is necessary. Perhaps choose one parameters and talk about that one in the paper. The rest could go in the supplemental materials.

**This figure was reconfigured to only include Observed OTUs, Chao1 and Shannon indices for simplicity (Fig. 4A). The main text (Section 3.3) and figure caption was modified to reflect these changes.**